# A Unified Hallucination Mitigation Framework for Large Vision-Language Models

**Yue Chang**[*]  **Liqiang Jing**[*]  **Xiaopeng Zhang**[*]  **Yue Zhang**

*The University of Texas at Dallas*
*{cjcy89253033,jingliqiang6,zhangxiaopeng0721}@gmail.com, yue.zhang@utdallas.edu*

**Reviewed on OpenReview:** *https://openreview.net/forum?id=ZVDWzgk6L6*

## Abstract

Hallucination is a common problem for Large Vision-Language Models (LVLMs) with long generations which is difficult to eradicate. The generation with hallucinations is partially inconsistent with the image content. To mitigate hallucination, current studies either focus on the process of model inference or the results of model generation, but the solutions they design sometimes do not deal appropriately with various types of queries and the hallucinations of the generations about these queries. To accurately deal with various hallucinations, we present a unified framework, **Dentist**, for hallucination mitigation. The core step is to first classify the queries, then perform different processes of hallucination mitigation based on the classification result, just like a dentist first observes the teeth and then makes a plan. In a simple deployment, Dentist can classify queries as perception or reasoning and easily mitigate potential hallucinations in answers which has been demonstrated in our experiments. On MMbench, we achieve a 13.44%/10.2%/15.8% improvement in accuracy on Image Quality, a Coarse Perception visual question answering (VQA) task, over the baseline InstructBLIP/LLaVA/VisualGLM.

## 1 Introduction

Hallucination in Large Vision-Language Models (LVLMs) is a critical issue, which manifests as the model's generated content partially deviating from the actual content of the image (Jing et al., 2023a). For example, when provided with a image and two questions as input, LVLMs inaccurately identify characters' actions and misinterpret relationships between characters, as illustrated in Fig. 1. Such inaccuracies can lead to misinformation, potentially degrading the user experience and misleading individuals. This issue underscores the necessity for ongoing improvements to enhance the reliability and accuracy of LVLMs, mitigating the risk of hallucinations and their consequent misinformation.

To tackle the above challenge, existing work either focuses on optimizing the training data and the parameters of the existing model (Liu et al., 2023b; Lu et al., 2023), or correcting the hallucinations during the generation stage without model update (Yin et al., 2023). The former collects high-quality training data, such as adding negative instances to the training data to avoid overconfidence in the model (Liu et al., 2023a). The latter mainly utilize the generated object information from the vision foundation model (such as blip2 (Li et al., 2023a)) to detect hallucinations and eliminate them. For example, Woodpecker (Yin et al., 2023) extracts main objects from the response generated by LVLMs and then verified these objects with object segmentation tool (Liu et al., 2023d) and VQA models (Li et al., 2023a). Similarly, HalluciDoctor (Yu et al., 2023a) makes the description-oriented answer chunks extraction and formulates corresponding questions, uses answers for these questions which are gathered from various LVLMs to do the consistency cross-checking and remove hallucinations.

---

[*]Equal contribution.

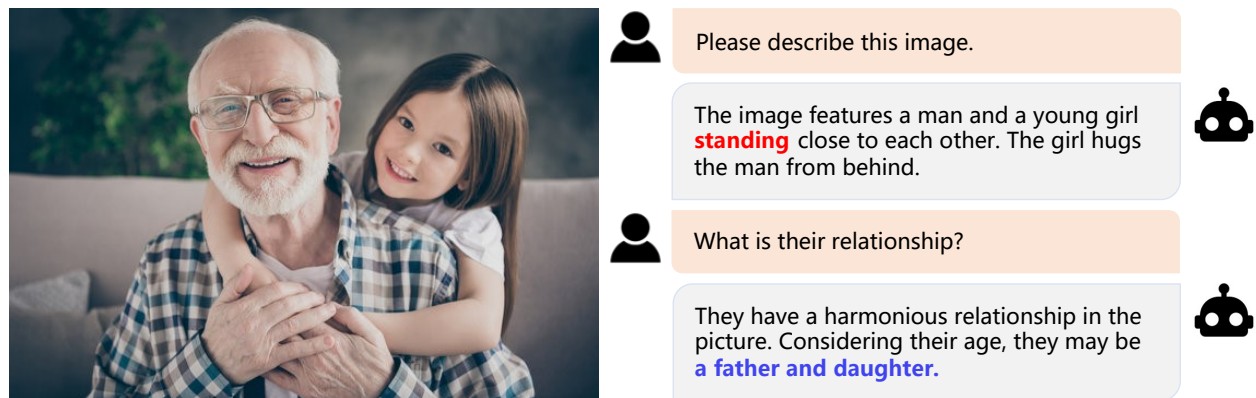

Figure 1: An example image of hallucination. The generation of the model is partially inconsistent with the image, which we call perception hallucination and reasoning hallucination respectively.

There are two main hallucinations in the model response (Ji et al., 2023): perception hallucination and reasoning hallucination, as shown in Fig. 1. The former is manifested by incorrectly describing image content in the model generation, such as errors when describing object attributes, while the latter refers to the model producing fallacies in logical reasoning answers. Although the previous methods for hallucination mitigation have achieved success, they still have a common problem, that is, when faced with these two types of hallucinations, the fixed verification method may sometimes be inappropriate and ineffective. For example, for reasoning queries, it is not effective to use object detection on pictures to verify whether the object in the answer exists, as shown in Fig. 1 In addition, sometimes when the corrected answer obtained by the existing method is used as input to perform the same correction once, the answer after the second correction is inconsistent with the first time, which means that one correction did not remove all the hallucinations.

In order to solve this problem, we propose a unified framework for two main kinds of hallucination mitigation. Whether it is a descriptive answer or a logical reasoning answer, our framework Dentist will try to correct the parts of the answer that do not match the content of the picture. Specifically, the framework we proposed is a verification loop, and each loop is divided into two core steps: (1) *Potential hallucination classification* divides the query into two categories: perception and reasoning, which also classifies the potential hallucinations in answers when these queries are used as input to LVLMs. (2) *Divide-and-conquer treatment* makes the mitigation based on the classification. The generation for the perception query will be verified by the sub-questions, while the generation for the reasoning query will be verified with the help of Chain-of-Thought (CoT). To ensure that hallucinations are mitigated as much as possible, the above verification loop will continue until the revised generation no longer changes semantically significantly or the loop limit is reached.

We complete quantitative experiments on MMbench (Liu et al., 2023e), LLaVA-QA90 (Liu et al., 2023c), CHAIR (Rohrbach et al., 2018) and POPE (Li et al., 2023c) using three models: InstructBLIP (Dai et al., 2023), VisualGLM (Ding et al., 2021; Du et al., 2022) and LLaVA (Liu et al., 2023c), respectively, to test the effectiveness of our proposed method. In addition, we also conduct a comparative experiment with a current effective LVLM hallucination mitigation method, Woodpecker (Yin et al., 2023). Our method demonstrates the effectiveness and superiority in many visual language tasks, and promotes the performance of the baseline models. In particular, in the experiment, we achieve a 13.44%/10.2%/15.8% improvement in accuracy in the visual language task of Image Quality, compared with the baseline models InstructBLIP/LLaVA/VisualGLM.

Our main contributions are summarized as follows:

- We propose a unified framework called Dentist for hallucination classification and mitigation. To the best of our knowledge, we are the first to distinguish treatment based on classification of the potential hallucinations and moreover use a validation loop for complete removal of hallucinations.

- Our unified framework is easily integrated into various LVLMs. The clear design of the framework also provides convenience for new classifications and treatments to access the framework.

- We comprehensively evaluated our method on several hallucination mitigation benchmarks (including MMbench, POPE, CHAIR, and LLaVA-QA90) with a detailed analysis. As a byproduct, we released our code[1].

## 2 Related Work

### 2.1 Large Vision-Language Model

Inspired by the success of Large Language Models (LLMs) (Wang et al., 2022; Zhao et al., 2023; Jing et al., 2024a; 2023b), the multimodal learning community shifted research attention to LVLMs. LVLMs mainly use the cross-modality aligner to connect the visual encoder (such as CLIP (Radford et al., 2021)) and LLMs (such as LLaMA (Touvron et al., 2023)) to tackle vision-language tasks. For example, LLaVA (Liu et al., 2023c) connects a vision encoder and a LLM for general-purpose visual and language understanding, suggesting practical tips for building a general-purpose visual agent. Meanwhile, InstructBLIP (Dai et al., 2023) introduce an instruction-aware Query Transformer which extracts visual features from the output embeddings of the frozen image encoder, and feeds the visual features as soft prompt input to the frozen LLM. In addition, VisualGLM (Ding et al., 2021; Du et al., 2022) use Qformer (Li et al., 2023b) which builds a bridge between the visual model and the language model. Though these LVLMs have powerful visual language understanding ability on the generation task, sometimes their outputs still contain hallucinations that need to be corrected. Some studies (Wang et al., 2023b; Awal et al., 2023) use LLMs to improve the performance of LVLMs, which is worth learning from.

### 2.2 Hallucination

With the progress of research on LVLMs, the problem of hallucination has gradually been exposed (Bai et al., 2024), and it has attracted more and more attention. Research around hallucination focuses on three aspects: detecting (Gunjal et al., 2023; Luo et al., 2024), mitigating (Kang et al., 2023; Lu et al., 2023; Wang et al., 2023a; Yin et al., 2023; Yu et al., 2023a;b; Leng et al., 2023; Huang et al., 2023; Jing & Du, 2024), and evaluating hallucinations (Jing et al., 2023a; Wu et al., 2024; Jing et al., 2024b). In this paper, we mainly focus on hallucination mitigation. Previous works on hallucination mitigation can be divided into two categories: model inference optimization and model generation optimization. The first category focuses on the process of the training and inference of the LVLMs. RLHF-V (Yu et al., 2023b) collects human feedback at the data level and learns the correctional human feedback at the training level to reduce hallucinations in model generations. Ever (Kang et al., 2023) points out that mitigating hallucinations in real time during model inference is more appropriate than generating corrections from the model outputs, as the latter is subject to snowballing effects. VIGC (Wang et al., 2023a) uses an iterative method to concatenate the short sentences generated each time, and ensures accuracy by controlling the length of the generation. On the other hand, the second category focuses on the aspect of the generation of LVLMs, designing methods to obtain hallucination information from the output of the model and do the mitigation. Leng et al. (2023) introduces Visual Contrastive Decoding (VCD) to counteract the statistical biases and mitigate hallucinations by contrasting model outputs generated from original and distorted visual inputs. For example, Woodpecker (Yin et al., 2023) makes the question formulation and visual knowledge validation base on the keywords which are extracted from the output of the model and uses an LLM to modify the hallucinations in the generated responses.

Despite the success of the existing method, they overlook the diversity of hallucinations which results in a fixed hallucination elimination method that cannot be applied to all hallucination situations well. To solve this problem, we propose a unified framework for mitigating hallucinations, the core step of which is to classify potential hallucinations caused by different queries.

---

[1]https://github.com/CYandYue/Dentist.

# 3   Method

In order to tackle various types of hallucination, our objective is to propose a unified framework for hallucination mitigation. Therefore, we devise a unified hallucination mitigation framework for LVLMs which mainly consists of three major components: potential hallucination classification, divide-and-conquer treatment, and validation loop, as shown in Fig. 2.

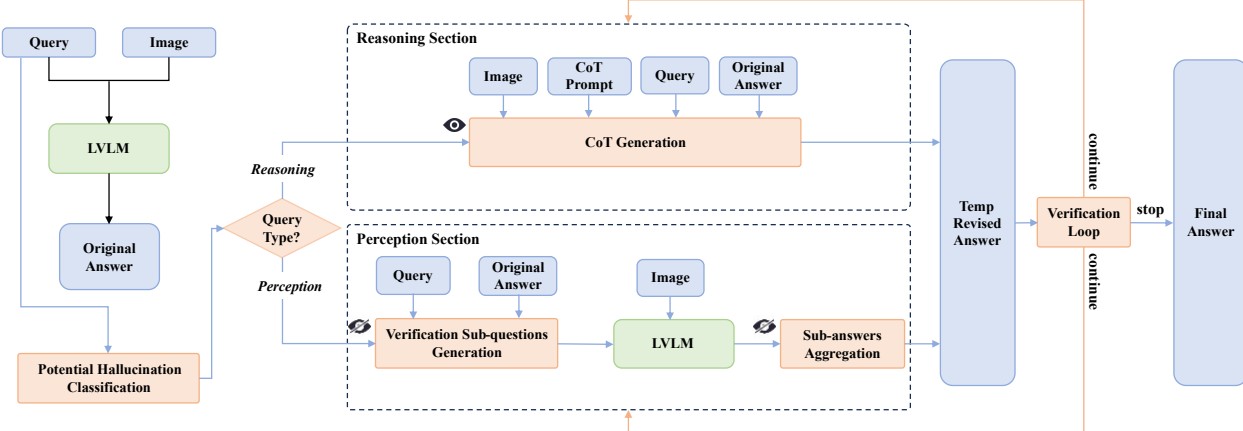

Figure 2: An overview of the proposed method. The components using GPT are indicated in orange. The icons of open and closed eyes indicate whether the component is a pure text task or is related to an image. The black line represents the original part of LVLM. The blue line represents the forward path of the verification process, and the orange line represents the feedback path in the verification loop. The core point is to customize different methods of mitigating hallucinations by classifying the query. The reasoning section is used to mitigate the hallucinations caused by reasoning queries, while the perception section is used to mitigate the hallucinations caused by perception queries.

## 3.1   Task Formulation

Suppose we have a dataset $\mathcal{D} = \{(Q_i, I_i, Y_i)\}_{i=1}^{N}$, where $I_i$ is the image, $Q_i$ refers to the query of the image and $Y_i$ represents the truth answer to the corresponding query. Note $i$ represents the index of samples in the dataset $\mathcal{D}$. We omit the index of $Q_i, I_i$ and $Y_i$ in the following discussion for the sake of brevity. Thereafter, we feed the image and query sample to the LVLM as follows,

$$\hat{Y} = M(I, Q|\Theta), \tag{1}$$

where $\hat{Y}$ is the original answer for $(Q, I)$ and $\Theta$ is the parameters of LVLM $M$. Since the generated response $\hat{Y}$ suffers from hallucination problem, our aim is to devise a hallucination mitigation method to minimize the semantic difference between $\hat{Y}$ and $Y$ as follows,

$$\min(D(Y, R(\hat{Y}))), \tag{2}$$

where $D(\cdot)$ is the semantic deviation and $R(\cdot)$ refers to our hallucination mitigation method.

## 3.2   Potential Hallucination Classification

As we mentioned before, there are two main types of hallucination: perception hallucination and reasoning hallucination. These two types of hallucinations correspond to two types of queries: perception query and reasoning query (Liu et al., 2023e). The perception query feature mainly requires the model to have the ability to perceive visual features, such as attribute recognition, scene description, etc. The hallucinations caused by this type of queries can be effectively alleviated through visual level approaches, like object detection. The

reasoning query mainly tests the understanding and reasoning ability of the model and the corresponding potential hallucinations should be mitigated by reasoning level approaches such as CoT which significantly improves the ability of large language models to perform complex reasoning (Wei et al., 2022). Since the type of potential hallucination in the answer can be judged based on the type of query, we firstly classify queries into the above two major categories and then handle the corresponding potential hallucinations. We employ ChatGPT to complete query classification through a prompt as follows,

$$C = ChatGPT(P_c(Q)), \tag{3}$$

where $P_c(\cdot)$ is a prompt which can instruct ChatGPT to classify the query into perception or reasoning, and $C$ represents the classification result. Corresponding details can be found in Appendix A.3.

### 3.3 Divide-and-conquer Treatment

After query classification, the LVLM responses of perception and reasoning queries need to be processed differently. This is because, as we mentioned before, different types of queries examine different capabilities of the LVLM, and the mitigation methods required for hallucinations in answers are also different.

To deal with the perception query, we need to generate verification sub-questions based on the original query and the original answer with hallucinations generated by the LVLM. The LVLM answers these sub-questions to obtain sub-answers and finally we aggregate these answers to form the output with fewer hallucinations. For reasoning queries, a common phenomenon is that the LVLM only generates the results of logical reasoning and the logical reasoning process we want to see is omitted in the generated content. In response to this situation, the method we propose is to use the CoT.

### 3.3.1 Visual Verification for Perceptron

LVLMs are prone to hallucinations when generating long descriptive texts (Liu et al., 2023b). This exactly corresponds to the situation of the responses of perception queries. We are inspired that when the long answer to perception queries contains hallucinations, we can split the long answer into short sentences and design verification sub-questions based on the key points in the sentences. We formulate this process as follows:

$$\{q_1...q_n\} = ChatGPT(P_s(Q, \hat{Y})), \tag{4}$$

where $P_s(\cdot)$ is a prompt which can instruct ChatGPT to generate sub-questions $q_i$ according to the query $Q$ and the original answer $\hat{Y}$, and corresponding details can be found in Appendix A.4.

After generating verification sub-questions, we feed them to the LVLM along with the original image. Thereafter, we can get the verification sub-answers from LVLMs. It is worth mentioning that the LVLM we used for generating the sub-answers is just the original model which has hallucinations that need to be revised. It can probably be replaced with any visual question answering (VQA) model, but would be accompanied by the suspicion of using a better model for better work. To demonstrate the ability of our approach to mitigate hallucinations rather than the ability of the rectified models, we chose to use the original LVLM. We formulate this process as follows:

$$y_i = M(I, q_i), \tag{5}$$

where $y_i$ is the $i$-th sub-answer.

Then, we aggregate the sub-question-answer pairs and feed them to ChatGPT with the original answer $\hat{Y}$ to refined the hallucinated response as follows,

$$\overline{Y} = ChatGPT(P_a(\{q_1, y_1\}...\{q_n, y_n\}, \hat{Y})), \tag{6}$$

where $P_a(\cdot)$ is a prompt that can instruct ChatGPT to aggregate the sub-answers and do the correction, and $\overline{Y}$ is the corrected answer. Details for the prompt $P_a(\cdot)$ template which is used for aggregating sub-answers can be found in Appendix A.5.

### 3.3.2 Chain of Thought for Reasoning

The answers generated by the LVLM from reasoning queries are not as "clear" as the answers to perception questions, which means the LVLM's answers tend to only contain the results of logical reasoning, but not the process of logical reasoning and the basis for that (perception about visual details before reasoning). Therefore, the method of the perception section is no longer applicable because of the missing part about perception in the reasoning answer. To solve this problem, we use the CoT prompt method to obtain answers that contain more details which are beneficial to our following process. At the same time, the LVLM will also improve the accuracy of reasoning when performing CoT. Add "Let's think step by step" to the start of the original query to do the CoT and use ChatGPT by prompt to obtain the revised answer as follows:

$$\overline{Y} = ChatGPT(P_r(M(I, P_t(Q)), \hat{Y})) \tag{7}$$

$P_t(\cdot)$ refers to "Let's think step by step" and $P_r(\cdot)$ is the prompt that can instruct ChatGPT to correct the original answer $\hat{Y}$ according to the generation of LVLM $M$ with CoT. Refer to Appendix A.6 for the detailed information about prompt $P_r(\cdot)$ template.

### 3.4 Validation Loop

After the above steps, we obtain the preliminary verified answer $\overline{Y}$ which may still contain hallucinations that have not been eliminated because of the imperfections of the verification sub-questions generation. In order to solve this problem, we propose to regard the entire verification framework as a repeated block in the verification loop chain. We show the overall procedure in Algorithm 1. The verified answer is treated as the original answer and re-verified. The difficulty of loop verification is how to judge when the hallucinations in the answer has been completely removed so the loop can be stopped. We believe that if and only if the verified answer does not change significantly semantically after a new round of verification, it means that all the hallucinations that can be eliminated have been eliminated. On the other hand, if the answer still changes significantly after a specific number of rounds of verification, we believe that there is a snowball error phenomenon in the verification cycle. We will stop the loop and use only the answer from the first verification as the final revised answer. We use ChatGPT to determine whether the answer has converged and is no longer changing semantically. This corresponds to the Similar function in Algorithm 1 and the prompt template is given in Appendix A.7.

---

**Algorithm 1** Dentist

---

**Input:** Original question Q, original image I, original answer $\hat{Y}$, the large vision-language model $M$, the maximum iteration $T$
**Output:** Corrected answer $\overline{Y}$
1: $Y_i \leftarrow \{\}$
2: $Y_l \leftarrow \hat{Y}$
3: **for** j in 1,2...$T$ **do**
4:     $Y_t \leftarrow Verify(Q, I, Y_l, M)$
5:     **if** j = 1 **then**
6:        $Y_i \leftarrow Y_t$
7:     **end if**
8:     **if** $Similar(Y_l, Y_t) \rightarrow Yes$ **then**
9:        # No improvement in new round of verification
10:        **return** $Y_l$
11:     **else**
12:        $Y_l \leftarrow Y_t$
13:     **end if**
14: **end for**
15: # Arrive the maximum iteration, so return $Y_i$
16: **return** $Y_i$

---

## 4 Experiment

In this section, we conduct extensive experiments to answer the following research questions:

**RQ1**. Could our framework improve the current LVLMs?

**RQ2**. What is the contribution of each component of our Dentist ?

**RQ3**. What is the intuitive performance of our Dentist ?

### 4.1 Experiment Settings

**Benchmarks. MMBench** is a novel multi-modality benchmark, which develops a fine-grained ability assessment for LVLMs. The MMBench evaluation standard is divided into three levels. The L-1 ability level incorporates Perception and Reasoning, L-2 ability level consists of Coarse Perception, Fine-grained, etc. and L-3 ability level covers Image Style, Image Scene, Image Emotion, etc. Relying on such hierarchical and fine-grained capability assessment, the performance of LVLM can be comprehensively evaluated. The dataset we use is MMBench-Test(EN).

**LLaVA-QA90** is also a dataset used to evaluate LVLMs. LLaVA-QA90 contains 90 questions and 30 images taken from COCO Val 2014 (Lin et al., 2014). To evaluate the generated response, we feed the query, image, and model response to GPT-4V (OpenAI, 2023) to get a score of a scale of 1 to 10. The prompt template is available in Appendix A.9. We respectively pair baseline LVLMs with Dentist, baseline LVLMs with Woodpecker, and provided their responses to GPT-4V for scoring, which ensures that the scores are mutually referenced, thereby making them more reliable.

**Caption Hallucination Assessment with Image Relevance (CHAIR)** (Rohrbach et al., 2018) is a widely-used metric for evaluating object hallucination in image captioning tasks. By comparing the image captions generated by the model with the ground truth objects in the corresponding image, CHAIR evaluates the degree of hallucination of the model and explains the performance of the model. CHAIR has two variants: $CHAIR_s$ ($C_s$) and $CHAIR_i$ ($C_i$), both of which reflect the degree of hallucination of the model, the difference being that $CHAIR_s$ is at the sentence level and $CHAIR_i$ is at the object instance level. The calculation is as follows:

$$CHAIR_s = \frac{|\{sentences\ with\ hallucinated\ object\}|}{|\{all\ sentences\}|}$$

$$CHAIR_i = \frac{|\{hallucinated\ objects\}|}{|\{all\ mentioned\ objects\}|}$$

**POPE** (Li et al., 2023c) is also an evaluation method for object hallucination. Three kinds of sampling settings of random, popular, adversarial, are constructed on the dataset according to human annotation or automatic visual segmentation tools. The difference between them lies in the negative sample sampling method. Where, the random setting randomly samples objects that do not exist in the image; the popular setting samples objects that are not present in the current image, but are most common throughout the dataset; the adversarial setting samples objects in the dataset that co-appear most frequently with objects in the current image, but are not present in the current image. POPE has a high degree of fairness and robustness, which helps us to better demonstrate the hallucination mitigation effect of Dentist.

In terms of sampling settings, we sample 100 images and construct 6 questions for each type of sampling setting for each image. Each question is a "Yes or No" question that transforms the model task into a simpler binary classification task. In terms of evaluation metrics, we adopt Accuracy, Precision, Recall and F1 score as the evaluation metrics.

**Baselines.** We first select 3 currently mainstream LVLMs as our baseline models, including **InstructBLIP**, **LLaVA**, and **VisualGLM**. In addition, we also compare against the baseline Woodpecker (Yin et al., 2023) which is a training-free hallucination correction method for LVLMs.

Table 1: Results on MMBench-Test(EN). The performance is measured by accuracy, where the improved performance for each partition is highlighted in bold.

| Ability | | InstructBLIP-7B | | | LLaVA-V1.5-7B | | | VisualGLM-6B | | |
|---|---|---|---|---|---|---|---|---|---|---|
| | | Baseline | Dentist | Woodpecker | Baseline | Dentist | Woodpecker | Baseline | Dentist | Woodpecker |
| Coarse Perception | Image Topic | 60.0 | 60.0 | 43.5 | **97.6** | 96.4 | 87.1 | 52.9 | 50.2 | 64.7 |
| | Image Quality | 0.0 | 10.6 | 0.0 | 0.0 | 10.2 | 0.0 | 0.0 | **15.6** | 3.5 |
| | Image Emotion | 32.7 | 41.3 | 12.0 | 67.5 | **76.4** | 67.5 | 41.7 | 50.6 | 33.7 |
| | Image Scene | 58.1 | 60.3 | 57.4 | 85.3 | **88.3** | 79.8 | 68.5 | 70.3 | 59.7 |
| | Image Style | 38.8 | 37.1 | 38.8 | **58.8** | 55.3 | 58.8 | 30.6 | 35.8 | 24.7 |
| Fine-grained Perception [Single-instance] | OCR | 51.9 | 58.6 | 36.4 | 70.1 | **78.3** | 67.5 | 41.6 | 43.6 | 53.2 |
| | Celebrity Recognition | 40.8 | 49.2 | 68.6 | 60.2 | **68.6** | 60.2 | 52.5 | 55.2 | 46.6 |
| | Object Localization | 3.9 | 14.4 | 8.6 | **16.3** | 11.5 | 14.4 | 8.6 | 10.9 | 8.6 |
| | Attribute Recognition | 46.5 | 51.4 | 52.5 | 66.7 | **70.6** | 66.7 | 40.0 | 43.7 | 33.3 |
| Fine-grained Perception [Cross-instance] | Action Recognition | 58.5 | 57.5 | 20.4 | **87.5** | 85.2 | 80.7 | 35.2 | 38.6 | 28.4 |
| | Attribute Comparison | 2.6 | 2.6 | 0.0 | 21.2 | **25.8** | 6.4 | 8.8 | 10.8 | 6.4 |
| | Spatial Relationship | 11.1 | 8.6 | 11.1 | 11.1 | **15.3** | 11.1 | 7.3 | 10.9 | 8.6 |
| Attribute Reasoning | Identity Reasoning | 68.3 | 68.3 | 70.7 | 86.6 | **86.6** | 86.6 | 81.7 | 88.4 | 71.2 |
| | Function Reasoning | 46.2 | 49.6 | 50.9 | 74.5 | **77.8** | 73.6 | 44.9 | 50.6 | 37.7 |
| | Physical Property | 21.0 | 21.9 | 26.0 | **55.0** | 50.0 | 53.0 | 26.0 | 30.3 | 17.0 |
| Relation Reasoning | Nature Relation | 22.2 | 27.3 | 24.7 | 38.3 | **38.3** | 38.3 | 24.7 | 30.6 | 8.6 |
| | Physical Relation | 11.3 | 17.3 | 19.2 | 28.9 | **28.9** | 26.9 | 3.8 | 9.6 | 3.8 |
| | Social Relation | 27.6 | 41.0 | 38.4 | 62.8 | **69.6** | 62.8 | 46.2 | 45.3 | 17.9 |
| Logic Reasoning | Image-Text | 5.9 | 7.0 | 15.8 | 11.9 | 10.9 | **12.9** | 3.9 | 5.0 | 8.9 |
| | Future Prediction | 46.7 | **55.0** | 25.0 | 43.1 | 52.1 | 44.4 | 21.6 | 31.0 | 6.9 |
| Overall | | 33.9 | 36.9 | 32.7 | 51.0 | **54.8** | 51.2 | 32.0 | 36.4 | 28.7 |

**Implementation Details.** We utilize GPT-3.5-turbo-0613[2] to assist in keyword extraction, sub-question generation, verification loop, and verification answer integration. Experiments have proven that GPT-3.5-turbo can tackle these tasks. On MMBench, we set the experiment rounds to 2: (1) In the first round of evaluation, we have the model generate raw predictions according to MMBench's evaluation rules and submit them to MMBench's official platform to obtain various accuracy rates; (2) In the second round of evaluation, based on the original prediction of the model, query classification, different verification processes and answer integration are carried out using GPT-3.5-turbo (specific details can be found in Section 3). Similarly, we upload the results of the second round of evaluation to the official MMBench platform to obtain various accuracy rates; (3) Finally, we jointly analyze the results of two rounds of evaluation to demonstrate the effectiveness and superiority of Dentist.

Previous studies (such as POPE (Li et al., 2023c)) have found a strong correlation between LVLMs hallucination and the length of the generated text, and in our experiments we find this to be true. In the CHAIR evaluation, since the LVLMs we select all have remarkable instruction following ability, we notice that when the LVLM are prompted to "generate as detailed a description as possible", the CHAIR score of the model is much higher than when they are prompted to "generate a short description" (a higher CHAIR score indicates a higher degree of hallucination). This is not desirable in a common usage scenario. But we can take advantage of this feature to better demonstrate the ability of Dentist to mitigate hallucinations. Therefore, in the CHAIR experiment, we prompt the model to generate a detailed description.

In the following experiments, we limit the maximum number of iterations to 3 (i.e., T in Algorithm 1 is equal to 3) to ensure the effectiveness of the verification and avoid excessive time costs.

## 4.2 Results (RQ1)

**Results on MMBench.** The results on MMBench are summarized in Tab. 1. From this table, we have several observations. (1) The largest accuracy improvement among the three LVLMs exceeds 15.6%, showing that Dentist have excellent correction effects, making obvious improvements in various metrics for the baselines. (2) Dentist performs outstandingly in Image Emotion, Image Quality, Future Prediction, Attribute Recognition, etc., which indicates that Dentist is capable of mitigating hallucination in coarse perception, fine-grained perception and logic reasoning. (3) Among all metrics, Image Quality shows the highest improvement, which indicates that Dentist is particularly effective for hallucinations in such problems.

---

[2]https://platform.openai.com/

Overall, Dentist has brought significant improvements to the three LVLMs. Comparing Woodpecker's performance, it can be seen that it can also bring many improvements in some perception queries, such as bringing a huge 11.8% improvement to VisualGLM on Image Topics. However, in many queries, especially reasoning queries, it even reduces LVLM's performance, such as causing a huge 28.3% decrease in VisualGLM on Social Relations, which is unacceptable. A large part of the reason for the decline in performance comes from Woodpecker's lack of targeted handling of reasoning problems.

Table 2: Results on LLaVA-QA90. The accuracy, detailedness and logicality metrics are on a scale of 10, and a higher score indicates the better performance. The better performance for each partition is highlighted in bold.

| LVLM | | Accuracy | Detailedness | Logicality | Precision (%) |
|---|---|---|---|---|---|
| InstructBLIP | Baseline | 6.5 | 4.9 | 4.3 | 21.0 |
| | Woodpecker | 6.4 | 5.5 | 4.4 | 20.9 |
| | Dentist | **7.0** (+0.5) | **5.5** (+0.6) | **4.8** (+0.5) | 22.8 |
| LLaVA | Baseline | 6.0 | 5.3 | 4.4 | 27.3 |
| | Woodpecker | 6.5 | 5.5 | 4.5 | 26.6 |
| | Dentist | **6.6** (+0.6) | **5.8** (+0.5) | **5.0** (+0.6) | 28.0 |
| VisualGLM | Baseline | 5.6 | 5.0 | 4.0 | 36.5 |
| | Woodpecker | 2.0 | 1.6 | 1.3 | 38.3 |
| | Dentist | **6.2** (+0.6) | **5.8** (+0.8) | **4.7** (+0.7) | 39.8 |

**Results on LLaVA-QA90.** If manual verification is required, the evaluation on LLaVA-QA90 is labor-intensive and somewhat subjective. Therefore, it is necessary to use a powerful evaluation tool to ensure consistency in evaluation standards while also possessing strong visual language task answering and instruction following abilities. Therefore, we consider utilizing the powerful LVLM, GPT-4V. Specifically, we involve GPT-4V in scoring and evaluating model responses by setting appropriate prompt words. We have designed the following three metrics: Accuracy: how accurate is the model response about the image content; Detailedness: level of details of the responses; Logicality: whether the reasoning content of response is reasonable. In addition, we also use GPT-3.5-turbo-0613 to calculate the proportion of logical reasoning sentences included in the passage without hallucination, and record it as Precision to better check the rationality of the reasoning content. The prompt template is available in Appendix A.9. We conduct the same evaluation on the current effective hallucination correction method, Woodpecker, and compare the baseline LVLMs, Dentist, and Woodpecker scores.

Tab. 2 shows the results. Obviously, equipped with our verification method, the models' performance has been comprehensively improved across the three metrics. On average, there is an improvement of over 0.5 points (relative improvement exceeding 13.6%), and Dentist scores better than Woodpecker on all baselines. This indicates that Dentist not only improves the accuracy and detailedness of LVLMs in describing image content, but also enhances the rationality of reasoning content. At the same time, the Precision of Dentist has also improved to some extent, indicating that Dentist can increase the proportion of effective reasoning content in LVLMs' answers.

It is worth noting that Woodpecker's score in VisualGLM decreases significantly. We find that the reason for the decrease in score is the failure of Woodpecker's object detector. On the one hand, this proves that the Woodpecker over-relies on its object detector, and when its detector fails, the correction effect of Woodpecker will become worse; on the other hand, it proves that Dentist has stronger robustness.

**Results on CHAIR.** We compare the effects of Dentist and Woodpecker on mitigating hallucinations. The specific method is as follows: we select 2000 examples in COCO Val 2014, let the baseline model generate corresponding captions, and then apply Dentist and Woodpecker respectively to process the hallucination correction of the captions, and calculate the two metrics of the CHAIR to analyze the difference between the two correction methods.

Tab. 3 shows the results. It is easy to draw the following conclusions from the results: (1) The model with remarkable performance may also produce more hallucinations. For example, LLaVA scored very high on

Table 3: Comparison of the CHAIR metrics between Dentist and Woodpecker. The best performance for each metric is in bold.

|  | InstructBLIP | | LLaVA | | VisualGLM | |
|---|---|---|---|---|---|---|
|  | $C_s \downarrow$ | $C_i \downarrow$ | $C_s \downarrow$ | $C_i \downarrow$ | $C_s \downarrow$ | $C_i \downarrow$ |
| Baseline | 59.2 | 22.9 | 84.0 | 23.4 | 44.0 | 17.8 |
| Woodpecker | 56.9 | 18.5 | **71.9** | 19.8 | 37.0 | 11.9 |
| Dentist | **52.1** | **16.7** | 75.2 | **17.1** | **36.0** | **11.3** |

MMBench, but the CHAIR evaluation shows that LLaVA's hallucinations were more serious. (2) Dentist demonstrates an ability to reduce hallucinations no less than Woodpecker, and helps the baselines reduce more hallucinations in most aspects.

Table 4: Results on POPE. The best performance for each metric is highlighted in bold.

| POPE | LVLM | Method | Accuracy | Precision | Recall | F1 Score | Yes (%) |
|---|---|---|---|---|---|---|---|
| Random | InstructBLIP | Baseline | 85.00 | 88.32 | 78.67 | 84.32 | 45.67 |
| | | Woodpecker | 86.21 | 92.38 | 78.97 | 84.63 | 35.08 |
| | | Dentist | **87.00** | **94.40** | **80.67** | **85.82** | 41.67 |
| | LLaVA | Baseline | 79.67 | 75.47 | 84.67 | 75.34 | 63.00 |
| | | Woodpecker | 84.78 | 85.64 | 90.26 | 83.44 | 50.23 |
| | | Dentist | **87.33** | **89.44** | **92.67** | **86.98** | 47.33 |
| | VisualGLM | Baseline | 53.44 | 51.85 | **99.12** | 68.09 | 95.79 |
| | | Woodpecker | 69.74 | 66.83 | 90.56 | 76.33 | 73.82 |
| | | Dentist | **75.88** | **73.25** | 87.79 | **78.95** | 68.02 |
| Popular | InstructBLIP | Baseline | 80.67 | 81.72 | **79.00** | 80.34 | 48.33 |
| | | Woodpecker | 81.03 | **88.81** | 68.97 | 78.43 | 37.93 |
| | | Dentist | **83.33** | 86.76 | 78.67 | **82.52** | 45.33 |
| | LLaVA | Baseline | 75.67 | 73.98 | **90.00** | 74.79 | 70.33 |
| | | Woodpecker | 85.42 | 84.38 | 85.39 | 83.64 | 50.28 |
| | | Dentist | **87.00** | **90.90** | 84.67 | **86.69** | 47.67 |
| | VisualGLM | Baseline | 58.31 | 54.63 | **99.12** | 70.44 | 90.90 |
| | | Woodpecker | 66.78 | 62.45 | 88.34 | 70.25 | 88.44 |
| | | Dentist | **70.53** | **68.12** | 87.35 | **71.20** | 86.91 |
| Adversarial | InstructBLIP | Baseline | 78.83 | 76.95 | **82.33** | 79.55 | 53.50 |
| | | Woodpecker | 77.59 | 82.23 | 68.97 | 75.47 | 41.38 |
| | | Dentist | **80.83** | **83.33** | 78.67 | **80.41** | 47.83 |
| | LLaVA | Baseline | 76.67 | 71.06 | **92.00** | 73.40 | 75.33 |
| | | Woodpecker | 77.62 | 74.53 | 88.67 | 80.73 | 64.25 |
| | | Dentist | **80.67** | **78.40** | 84.67 | **81.41** | 54.00 |
| | VisualGLM | Baseline | 54.10 | 52.21 | **99.12** | 68.40 | 95.12 |
| | | Woodpecker | 62.67 | 60.33 | 85.44 | 66.78 | 82.35 |
| | | Dentist | **68.10** | **63.42** | 86.90 | **68.87** | 80.90 |

**Results on POPE.** We evaluate InstructBLIP, LLaVA and VisualGLM for hallucination on POPE respectively, and compare the results of baseline LVLMs, Woodpecker and Dentist. Tab. 4 summarizes the results of POPE under the random, popular, and adversarial sampling settings. It can be seen that, in all sampling settings, VisualGLM is relatively weak, noting that its Recall and Yes Rate are both very high (close to 1), which indicates that VisualGLM produces relatively severe hallucinations of objects not in the image (i.e. negative samples). The reason for this phenomenon is probably that VisualGLM fails to deal with the imbalanced distribution of the dataset during the training process. The other baseline models are above 70% on these metrics. Dentist brings significant improvements to these baseline LVLMs, which validates Dentist's excellent performance in mitigating hallucination. Specifically, under the relatively simple

random sampling setting, Dentist obtains a gain of 22.44% for VisualGLM in terms of accuracy. In the more challenging popular and adversarial settings, the performance of these baseline LVLMs decline to varying degrees. Dentist continues to show remarkable performance. In particular, Dentist boosts the precision of LLaVA by 16.92% in the popular setting and accuracy of VisualGLM by 14.00% in adversarial setting. In addition, it can be seen from the comparison that Dentist outperforms Woodpecker in all sampling settings.

**Verification time comparison.** While evaluating Dentist and Woodpecker on POPE, we count and analyze the time spent on hallucination correction for both. A total of 100 images and 1,800 questions are provided to Dentist and Woodpecker throughout the evaluation process. For InstructBLIP, Dentist spend a total of 7 hours and 49 minutes to complete this task, with an average of 15.6 seconds per query. Woodpecker spend a total of 8 hours and 16 minutes to complete the task, with an average of 16.5 seconds per query. It can be seen that the the rapidity of Dentist is also better than that of Woodpecker and has a good timeliness.

**Comparison with Woodpecker.** The similarity between our framework Dentist and Woodpecker is that we both use an LLM to revise the hallucinated response generated by LVLMs. Woodpecker extracts main objects from responses and verifies these objects with object segmentation tool and VQA models, for all hallucinated responses. However, Dentist has a divide-and-conquer treatment. When Dentist is faced with perception query, it tries to verify the main objects in the model response. When faced with reasoning query, Dentist uses CoT to deal with it. This divide-and-conquer treatment accurately mitigates hallucinations and achieves better results.

### 4.3 Ablation Studies (RQ2)

To explore the effect of the query classification and verification loop, we conduct ablation studies in this section. **Query Classification.** We study three different variants and evaluate their performance on MMBench. (1) **w/o-Classification**: we disable the query classification section of Dentist ; (2) **w/o-Reasoning**: we classify all queries into perception for verification; (3) **w/o-Perception**: we classify all queries into reasoning for verification. For the sake of brevity, we will refer to these three variants as w/o-Cla, w/o-Rea and w/o-Per in the following discussion.

Table 5: Results on MMBench with different variants of InstructBLIP. For more comprehensive evaluation results on LLaVA and VisualGLM, please refer to the Appendix A.10.

| Variants | Perception Accuracy | Reasoning Accuracy |
|---|---|---|
| Baseline | 33.74% | 31.15% |
| Dentist | 37.62% | 35.92% |
| w/o-Classification | 34.86% | 32.73% |
| w/o-Reasoning | 38.94% | 25.48% |
| w/o-Perception | 28.34% | 38.44% |

Tab. 5 shows the results of InstructBLIP. We can see that: (1) If query classification is not performed and verification is performed directly (w/o-Cla), the accuracy is not much higher than the baseline, and in some cases there is even a problem of reduced accuracy. Because at this point, the way Dentist corrects the model's answers is completely random, which largely depends on the performance and habits of GPT-3.5-turbo: it can be seen that the perception accuracy may not differ much from the baseline, or slightly higher than the baseline, while the reasoning accuracy may decrease. This is because the query classification section tends to treat the problem as perception for processing. (2) If all queries are classified into perception (w/o-Rea) (this is what most current LVLM hallucination mitigation methods do), it can be seen that the perception accuracy is greatly improved, while the reasoning accuracy is greatly attenuated. This is because Dentist also verifies the reasoning problem as perception, so the verification method is not appropriate, resulting in a decrease in accuracy; (3) In the same way, if all problems are classified as reasoning (w/o-Per), the reasoning accuracy is greatly improved, and correspondingly, the perception accuracy is reduced; (4) It can also be found that the perception accuracy of w/o-Rea may even be slightly higher than that of Dentist. We speculate that this is due to the misjudgment by GPT-3.5-turbo when classifying queries, such as mistakenly categorizing a very small number of perception queries as reasoning, while w/o-Rea precisely corrects this part of the misjudged

perception queries. The same goes for reasoning queries. To verify this problem, we extract 1000 samples from the test results of MMBench and manually count the number of classification errors of Dentist. The results shows that among these 1,000 samples, 33 queries are misclassified, with an error rate of approximately 3.3%. Compared to its improvement in LVLMs performance, we believe this quantity is acceptable.

**Verification Loop.** Verification loop is also a component that we need to study. We conduct additional experiments by varying the number of verification loops in our framework and evaluating it on MM-Bench to demonstrate its effectiveness.

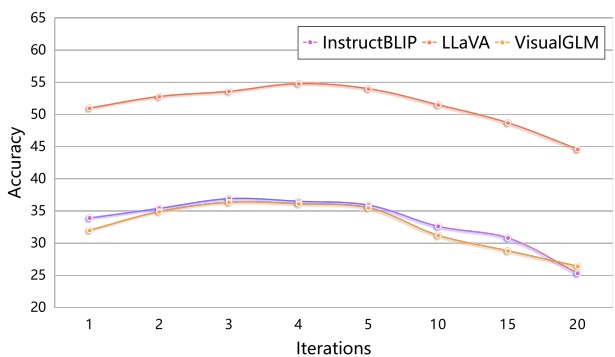

Fig. 3 shows the results. We can see that when the number of verification loop is small, there is a slight improvement in accuracy as the number of loops increases. However, when the number of cycles is large, the accuracy actually decrease as the number of cycles increases. We separately take out one of the cases for observation and found that when the number of cycles is large, the output of the LVLM and GPT gradually become chaotic and uncontrollable,

Figure 3: Results of verification loop

which may lead to an avalanche of decrease in the accuracy of the model when the number of cycles is large enough. Therefore, we conclude that verification loop is effective, but special attention needs to be paid to limiting its frequency. When the model answer matches the validation answer, it is important to exit the loop validation in a timely manner.

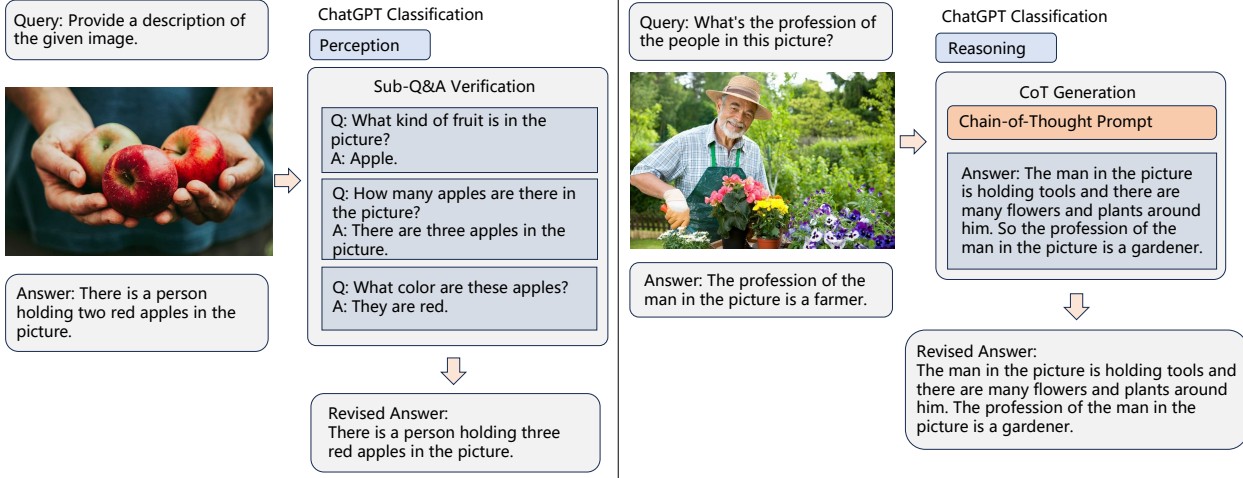

Figure 4: Two testing cases of Dentist . The left example shows a perception hallucination produced by LVLM, and the right example shows a reasoning hallucination produced by LVLM, and these hallucinations are eliminated by Dentist.

## 4.4 Case Study (RQ3)

We provide two testing examples in Figure 4 to conduct qualitative analysis. It is obvious from the above example that: In the first example, Dentist classifies the query as "Perception". The original response of LVLM is that "There is a person holding two red apples in the picture", which is obviously wrong. After Dentist extracts the keywords "apple", "two", "red", etc., three corresponding sub-questions are generated. Dentist then answers the sub-questions one by one. Since the sub-questions are more targeted and usually

have short outputs, the possibility of hallucinations is greatly reduced. After comparing and integrating the answers, Dentist finds out the existing hallucinations (the number of apples), corrects the original answer, and eliminates the perception hallucination.

In the second example, the original response of LVLM infers that the person in the image is a farmer, which is obviously wrong. The model mistakenly draws a conclusion that is contrary to the facts based on the information in the image. Dentist classifies the query as "Reasoning" and refines the hallucinated answer according to the content of the CoT and the original output of the model. Dentist matches the original answer and the CoT generation answer, and find that the objects and inference in the content of the two matches, so the final answer is obtained, thus eliminating the reasoning hallucination.

In the above cases, Dentist perfectly eliminates the perception and reasoning hallucinations produced by LVLM.

## 5 Limitations and Future Work

This study acknowledges limitation in the Dentist framework. When performing verification, we take the answers of the verification questions as the ground truth, which may still contain hallucinations. In terms of reasoning hallucination mitigation, the CoT for reasoning we use is relatively simple. In addition, loop verification also increases time cost. In future work, we may refine the CoT for reasoning and add validation of the details obtained from the CoT. In order to reduce time and money costs, simplifying prompts without compromising effectiveness is a feasible research direction.

## 6 Conclusion

In this work, we propose a unified framework for hallucination classification and mitigation. We are the first to distinguish treatment based on the classification of hallucinations and use a validation cycle for the removal of hallucinations. Our framework has a clear design which is easily integrated into various LVLMs, and provides convenience for new classifications and treatments to integrate into the framework. To evaluate the effectiveness of our framework, we conduct experiments on the three baseline LVLMs on MMbench, LLaVA-QA90, CHAIR and POPE, which shows that Dentist can significantly improve the baseline LVLMs on these benchmarks. At the same time, we compare the results of LLaVA-QA90 and CHAIR with those of Woodpecker, and the results shows that Dentist not only has excellent hallucination correction ability, but also has strong robustness.

### Acknowledgments

This work was supported by the OpenAI Research Access Program (Award ID: 0000006384), which provided access to advanced GPT models. The authors thank OpenAI for their support and for fostering innovative AI and machine learning research.

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

# A  Appendix

In this section, we will display all prompt templates used in this framework and other noteworthy figures and tables.

## A.1  Multiple Responses Study

In order to comprehensively study the robustness of Dentist and the consistency of experimental results, we re-conduct POPE evaluation under the random sampling setting, and the other settings are the same as those introduced in Section 4.1. The difference is that we let the LVLM test each example 10 times and let our framework process each of these responses. We provide two new baselines: (1) **Direct Rejection Baseline (DR Baseline)**: If all the responses of one sample have errors, deem the sample incorrect, else we randomly choose one of the responses which has no errors to be the answer. (2) **Repeated Correction Baseline (RC Baseline)**: For each example, when our framework detects hallucinations in all ten responses of the model, the hallucinations are corrected normally. We speculate that the performance of DR Baseline will significantly decrease because only the classification function of our framework are enabled to detect errors, while the answer correction function of our framework is disabled, and the performance of RC Baseline should be consistent with Tab. 4.

Table 6: Results on POPE under the random sampling setting.

| LVLM | Method | Accuracy | Precision | Recall | F1 Score | Yes Rate |
|---|---|---|---|---|---|---|
| InstructBLIP | DR Baseline | 79.17 | 81.81 | 75.00 | 78.26 | 38.33 |
| | RC Baseline | 85.56 | 92.48 | 82.12 | 81.37 | 40.22 |
| LLavA | DR Baseline | 71.70 | 71.93 | 74.55 | 73.21 | 44.34 |
| | RC Baseline | 87.33 | 86.97 | 91.38 | 82.79 | 46.41 |
| VisualGLM | DR Baseline | 50.83 | 51.79 | 92.06 | 66.29 | 89.16 |
| | RC Baseline | 76.88 | 74.35 | 86.54 | 78.18 | 63.25 |

From the table, we can draw the following conclusions: (1) The results are consistent with our predictions. Using our framework to classify and detect errors without correcting the answers leads to a significant drop in the performance of DR Baseline, while the performance of RC Baseline did not show much decline. DR Baseline show a drop of more than 10% in all metrics, especially the Accuracy of VisualGLM, which drops by 25.05%. (2) This shows that our framework is very effective in detecting hallucinations generated by models and correcting the model's answers.

## A.2   Discussion on Reproducibility

We provide detailed discussion on the effect of our framework on correcting hallucinations and the reproducibility of the above experimental results. It should be emphasized that since our method is training-free, all parameters of the model are fixed. We focus on the following questions: (1) When LVLM repeatedly generates captions for the same image, will it produce the same hallucinations? (2) When using our framework to process a series of model responses in (1), can we obtain consistent results? Can we guarantee that the hallucinations can be corrected every time?

We continue to discuss the results of Appendix A.1.From the DR Baseline, we can see whether the same LVLM will repeatedly hallucinate the same image, and from the RC Baseline, we can see whether our framework's correction of hallucinations is repeatable. We analyze the responses of LVLMs and the results in Tab. 6, and arrive at the following conclusions: (1) When the model parameters are fixed, it is very easy to hallucinate the same image repeatedly, as long as the same image and the prompt with the similar semantic are provided. This is also the reason for the poor performance of DR Baseline, as the same hallucinations repeatedly appears in multiple responses of LVLM. (2) Our framework is still able to find and correct the hallucinations generated by LVLM in the face of repeated tests, and the performance is almost the same as the previous experiment, that is, the performance of RC Baseline is not significantly deviated from Tab. 4 which shows that our framework is very effective in detecting hallucinations generated by models and correcting hallucinations. Therefore, we believe that our experimental results are reproducible and that the hallucination correction capability of our framework is reproducible.

## A.3   Query Classification

The prompt template is in Fig. 5.

## A.4   Sub-questions Generation

The prompt template is in Fig. 6.

## A.5   Sub-answers Aggregation

The prompt template is in Fig. 7.

## A.6   CoT Verification

The prompt template is in Fig. 8.

## A.7   Verification Cycle

The prompt template is in Fig. 9.

## A.8   Results on MMBench

The results on MMBench are in Fig. 12, Fig. 13 and Fig. 14.

### A.9    Prompt for GPT-4V-aided evaluation and GPT-3.5-aided precision calculation.

The prompt template for GPT-4V-aided evaluation is in Fig. 10, and the prompt template for GPT-3.5-aided precision calculation is in Fig. 11.

### A.10    Results of ablation study

The results of LLaVA and VisualGLM are in Table 8 and Table 9.

### A.11    Benchmark table

Benchmark details are in Table 10.

### A.12    Hallucination case

Cases of hallucination mitigation are in Fig. 15.

### A.13    ChatGPT cost

The cost of calling the GPT-3.5-turbo-0613 API is shown in Table 11.

We calculated the cost of calling the gpt-3.5-turbo-0613 API when evaluating on MMBench. In one round of evaluation, the total cost was about \$2.75, with an average cost of \$0.0004 per question.

### A.14    Model parameter

Details of LVLMs parameters are shown in Table 12.

### A.15    GPT-4V-aided evaluation alignment method

When evaluating on LLaVA-QA90, we respectively pair baseline LVLMs with Dentist, baseline LVLMs with Woodpecker, and provided their responses to GPT-4V for scoring. Thus the scores need to be aligned. The alignment of the scores is as followed: Suppose that when the responses of LVLMs and Dentist are provided to GPT-4V for scoring, their scores are $S_{Baseline-1}$ and $S_{Dentist}$ respectively, and when LVLMs is paired with Woodpecker, their scores are $S_{Baseline-2}$ and $S_{Woodpecker}$ respectively. The final aligned scores are shown in Table 7.

Table 7: The final aligned scores of GPT-4V-aided evaluation.

|  | Score |
| --- | --- |
| LVLMS | $S_{Baseline-1}$ |
| Dentist | $S_{Dentist}$ |
| Woodpecker | $S_{Woodpecker} \times S_{Baseline-1} \div S_{Baseline-2}$ |

Table 8: Results on MMBench with different variants of LLaVA

| Variants | Perception Accuracy | Reasoning Accuracy |
|---|---|---|
| Baseline | 53.52% | 50.13% |
| Dentist | 56.83% | 51.77% |
| Dentist/N | 54.39% | 48.76% |
| Dentist/P | 57.90% | 42.63% |
| Dentist/R | 50.43% | 52.11% |

Table 9: Results on MMBench with different variants of VisualGLM

| Variants | Perception Accuracy | Reasoning Accuracy |
|---|---|---|
| Baseline | 32.31% | 31.60% |
| Dentist | 36.35% | 36.35% |
| Dentist/N | 35.06% | 28.73% |
| Dentist/P | 37.83% | 22.60% |
| Dentist/R | 28.26% | 37.60% |

## Prompt

--------------------------------------------------------------------------------------------------

**Role:**
You are now one of my question classification assistants.
Please help me classify the question into two categories: perception or reasoning(binary classification).

**Rules:**
1.The classification result is only "perception" or "reasoning". Choose one of the two to output.
2.If the question focuses on perception ability, answer "perception"; if the question focuses on logical reasoning ability, answer "reasoning".
3.Don't answer anything else, your answer can only contain "perception" or "reasoning".

**Examples**:
 1.my input:  "How many people are there in this picture?"
   your answer: "perception"

 2.my input: "The person in the picture may do what soon?"
   your answer: "reasoning "

{add more examples}

Now please classify the following question according to the example and then answer "perception" or "reasoning":

Figure 5: Prompt template for classification

## Prompt

-------------------------------------------------------------------------------------------------

**Role:**
You are my language assistant for generating sub-questions.
Please generate sub-questions to verify the caption of the picture based on QA-examples below.

**Rules:**
1.The number of sub-questions cannot exceed three.
2.Extract keywords such as objects, quantities, and locations to generate sub-questions.
3.Each sub-question should have a different focus.
4.Don't ask repeated questions in different sub-questions.
5.If my input contains multiple choice questions, please generate sub-questions based on the question, options and answers.

**Examples**:
1.my input:
"Question: Write a detailed description for this picture.
Answer: The picture shows a man standing on the back of the yellow taxi, with a yellow shirt and black pants, and a blue backpack on his back. The taxi is driving on a city street with cars and taxis in the background."
sub-questions you generated:
"1.Is there a man standing on the back of a taxi in this picture?
2.What color are the T-shirt and pants that man wear?
3.What's in the background? "

{add more examples}

Now please generate verification sub-questions based on my input below:

Figure 6: Prompt template for generating sub-questions

Table 10: The number of questions in experiment

| Benchmarks | Questions | Tasks |
|---|---|---|
| LlaVA-QA90 | 90 | generative |
| POPE | 1800 | classification |
| MMBench | 6666 | classification |
| CHAIR | 2000 | generative |

Table 11: The cost of calling the GPT-3.5-turbo-0613 API

| Model | Price(input) | Price(output) |
|---|---|---|
| GPT-3.5-turbo-0613 | $1.5 / 1M tokens | $2.0 / 1M tokens |

Table 12: Details of the evaluated LVLMs.

| LVLM | Overall Parameters |
|---|---|
| InstructBLIP | 8B |
| VisualGLM | 8B |
| LLaVA | 7.2B |

## Prompt

------------------------------------------------------------------------------------------------

**Role:**
You are my language assistant for correcting or remaining my passage.
Below is a passage and some Q&A pairs. You need to modify the passage or just keep it unchanged based on the Q&A pairs.

**Rules:**
1.The information provided by the Q&A pairs is the ground truth, and the information in the passage may contain errors.
2.If the passage conflict with the Q&A pairs, find them and correct the passage based on the Q&A pairs. Try to make minimal changes to retain the original sentence. Then give me the passage which have been corrected by you.
3.If the passage has no confliction with the Q&A pairs, just keep the original passage and give me that.
4.At any time your output should only be a passage.

**Examples**:
1.Passage:"There are two apples in the picture, they look stale."
Q&A pairs:
Q:How many apples are there in the picture? A:There are three apples in the picture.
Q:Do these apples look fresh in the picture? A:No, they look stale.
Your output:"There are three apples in the picture, they look stale."

{add more examples}

Now I give you the passage and some Q&A pairs, please follow the examples and give me the passage you modified:

Figure 7: Prompt template for aggregating sub-answers to form the output after alleviating the hallucination

---

## Prompt
--------------------------------------------------------------------------------------------

**Role:**
You are my language assistant for correcting or remaining my passage.
Below are two passages.
**Rules:**
1.The second passage is the ground truth, the first passage may contain some errors.
2.If the first passage conflict with the second passage, find them and correct the first passage based on the second passage. Try to make minimal changes to retain the original sentence. Then give me the first passage which has been corrected by you.
3.If the first passage has no confliction with the second passage, just remain the first passage and give me that.
4.At any time your output should only be a passage.

**Examples**:
1.The first passage: " The profession of the man in the picture is a farmer."
The second passage: "The man in the picture is holding tools and there are many flowers and plants around him. So the profession of the man in the picture is a gardener."
Your output: "The profession of the man in the picture is a gardener."

{add more examples}

Now I give you two passages, please follow the rules and examples and give me your output:

---

Figure 8: Prompt template for CoT verification

---

## Prompt
--------------------------------------------------------------------------------------------

**Role:**
You are my language assistant for determining whether there is a conflict between two passages.

**Rules:**
1.Below are two passages.
2.If there is conflicting content between the two passages, you should answer "yes"
3.If there is no conflicting content between the two passages, you should answer "no"
4.At any time You can only answer yes or no.

**Examples**:
1.Passage 1: "There are two apples in the picture, they look stale."
Passage 2: "There are three apples in the picture, they look stale."
Your answer: "yes"

{add more examples}

Now I give you two passages, please follow the examples and give me your answer about whether there is a conflict between two passages.

---

Figure 9: Prompt template for judging when the validation cycle can be stopped

## Prompt

------------------------------------------------------------------------------------------------

You are my scoring assistant. You need to score two passages describing the picture based on the content of the picture.

What you need to pay special attention to is the hallucination, which refers to the conflict between the content of the passages and the content of the picture.

For example, the passage incorrectly describes the shape or color of the object in the picture, or makes wrong inferences based on the content of the picture.

Please rate the two passages on a scale of 1 to 10, where a higher score indicates better performance, according to the following criteria:

1. Accuracy: Refers to whether the description of the picture by the passage is accurate. Passages with fewer hallucinations should be given higher scores.

2. Detailedness: Refers to whether the description of the picture is detailed in the passage. Note that descriptions with hallucinations are not counted. Passages with more details should be given higher scores.

3. Logicality: Refers to whether the logical reasoning made by the passage based on the picture content is complex and reasonable. Note that the logical reasoning with hallucinations are not counted. Passages with more logical reasoning should be given higher scores.

Please output a single line for each criterion, containing only two values indicating the scores for Passage 1 and 2, respectively.

The two scores are separated by a space. Following the scores, please provide an explanation of your evaluation, avoiding any potential bias and ensuring that the order in which the responses were presented does not affect your judgment.

Passage 1:
{Original Answer}

Passage 2:
{Revised Answer}

Figure 10: Prompt template for GPT-4V-aided evaluation

**Prompt**

------------------------------------------------------------------------------------------------

You are my language assistant. You need to calculate the proportion of logical reasoning sentences included in the following two passages respectively.
Specifically, for each passage, you need to follow these steps to calculate the proportion:
1. Count the number of all sentences in the passage.
2. Count the number of logical reasoning sentences in the passage.
3. Divide the number of logical reasoning sentences by the number of all sentences to get the proportion.

Please output a single line, containing only two values indicating the proportion for Passage 1 and 2, respectively.
This means that you can only output two values and not any other text analysis. The two values are separated by a space.

Passage 1:
{Original Answer}

Passage 2:
{Revised Answer}

Figure 11: Prompt template for GPT-3.5-aided precision calculation

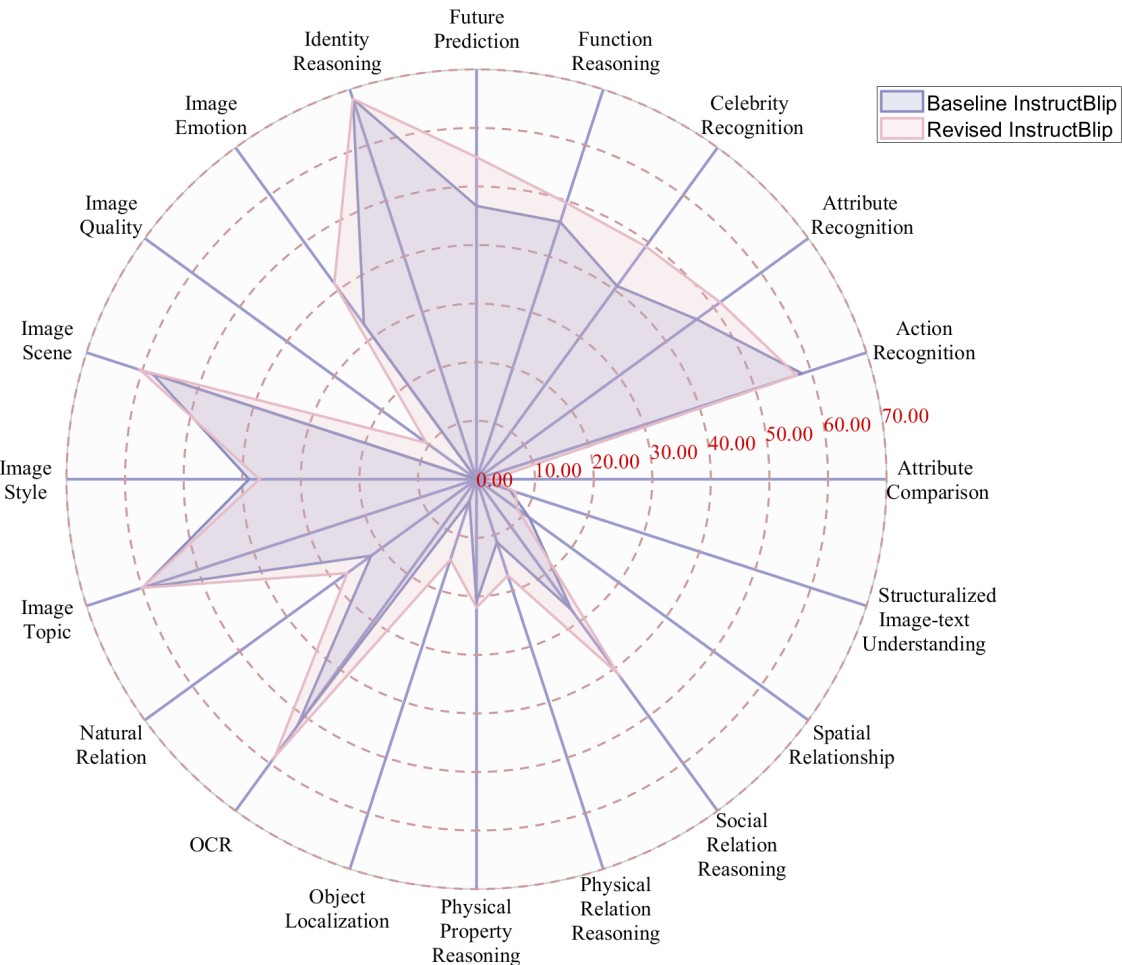

Figure 12: Results of InstructBLIP(Baseline and ours) across the 20 ability dimensions defined in MMBench. The blue area is the result of the baseline, and the red area is ours. See the legend. From this figure, we can intuitively see that our method can enhance the performance of baseline in terms of Image Impression, Image Quality and Future Prediction, etc. For more comprehensive evaluation results on LLaVA and VisualGLM, please refer to Fig.13 and Fig.14

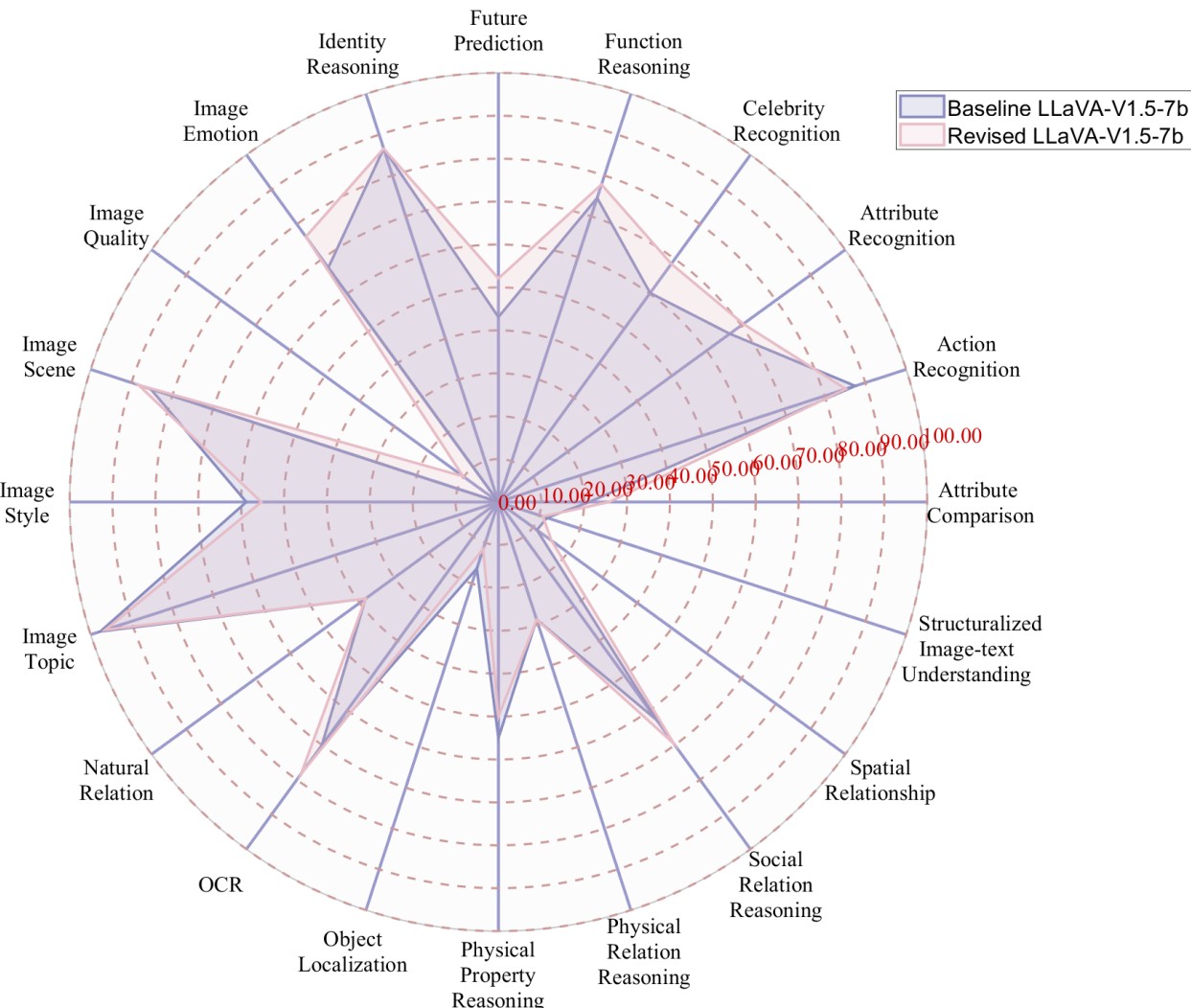

Figure 13: Results of LLaVA on MMBench.

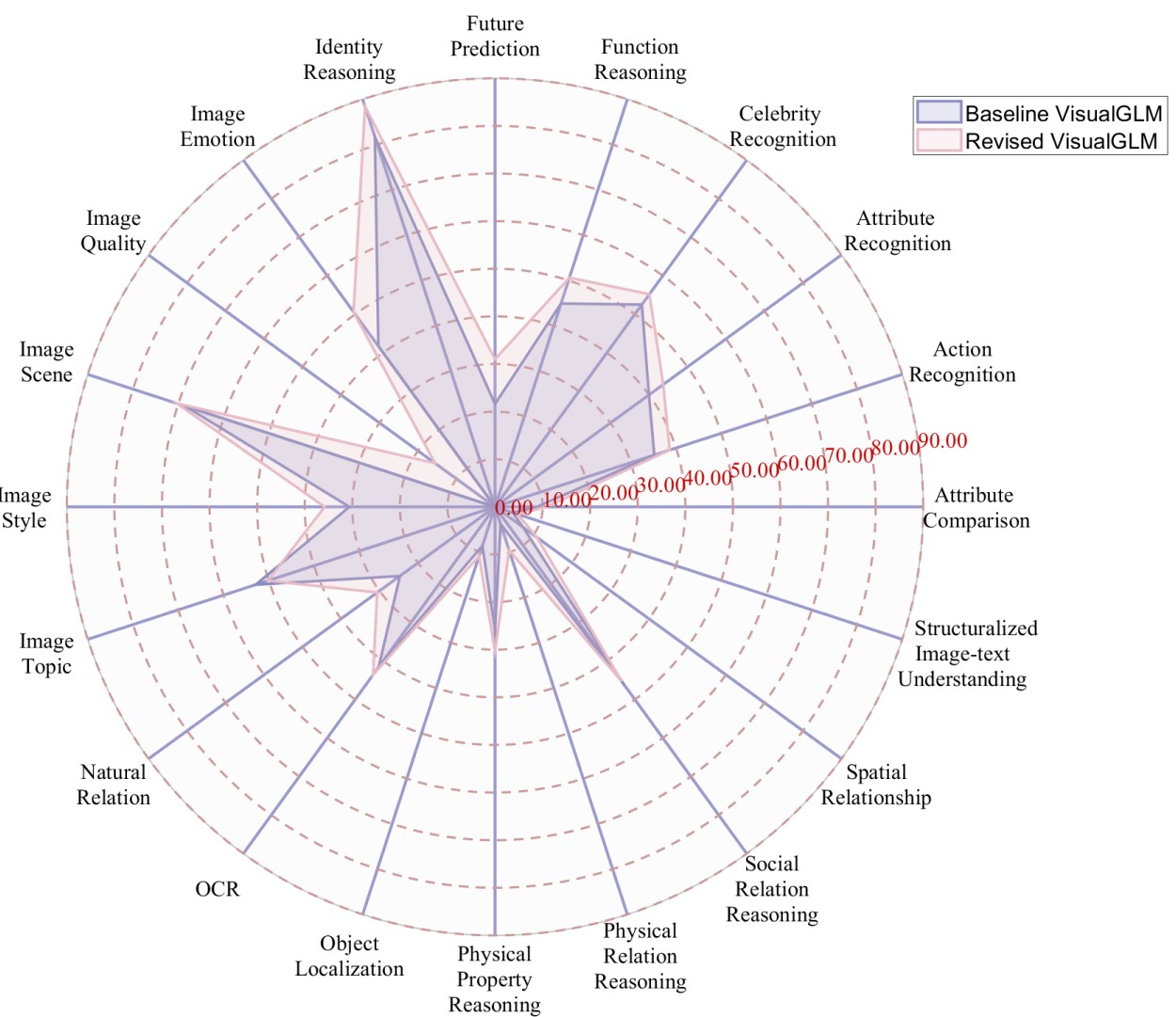

Figure 14: Results of VisualGLM on MMBench.

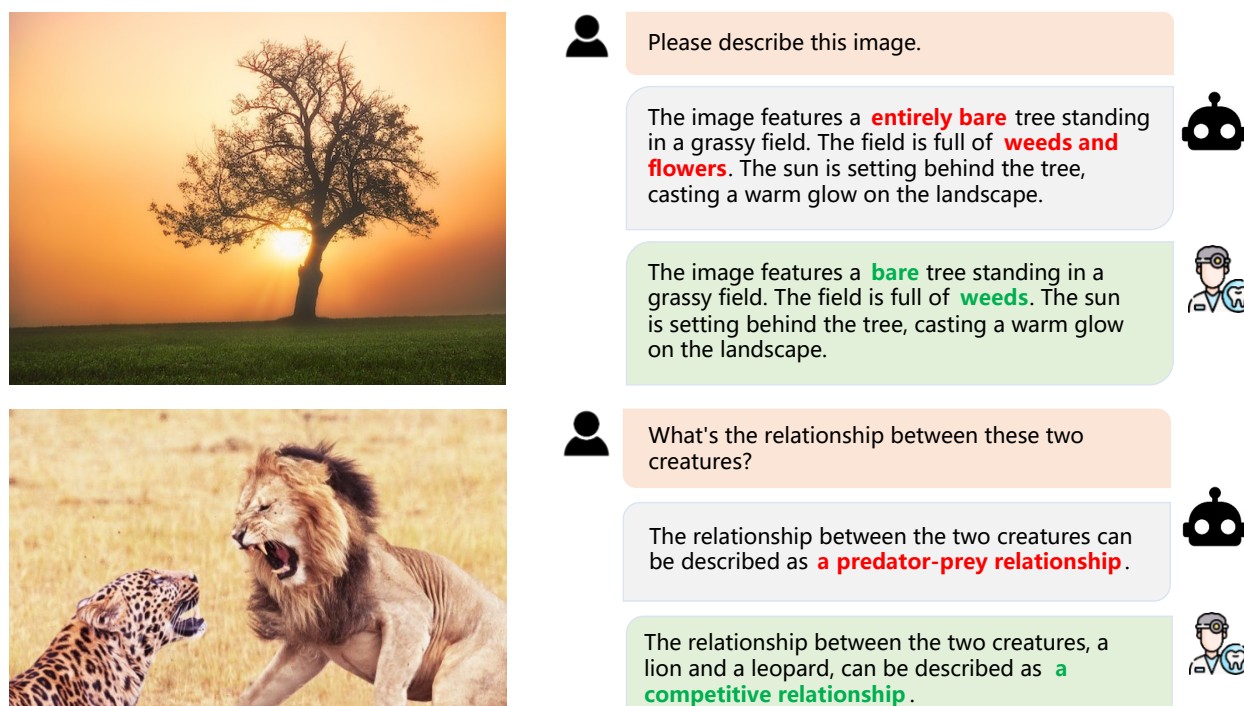

Figure 15: Example images of our hallucination mitigation. The part of the generation that conflicts with the content of the picture has been corrected.

