# OpenReview forum: "A Unified Hallucination Mitigation Framework for  Large Vision-Language Models"
_TMLR — Accepted by TMLR_

### Review · Reviewer_3BEp · 2024-05-31

**Summary Of Contributions:**

The authors introduce a prompting framework for mitigating hallucinations in VLLMs. They accomplish this by segregating reasoning errors from perception errors and then addressing them by querying either ChatGPT or a VLLM with revised prompts. They validate their approach on various visual understanding tasks (MMbench, POPE, CHAIR, and LLaVA-QA90) and present competitive baselines.

**Audience:**

Yes

**Claims And Evidence:**

Yes

**Requested Changes:**

Could you present an ensemble baseline where the VLLM is called multiple times, and for each generation, we leverage your classification to detect if there are any errors? For evaluation, if all the generations have errors, deem the sample incorrect, else randomly sample one of the generations where neither a perception nor a reasoning error is detected.

This baseline aims to understand if Dentist is the best framework for leveraging additional LLM/VLLM calls.

**Strengths And Weaknesses:**

Strengths:

1. The paper introduces a simple but effective framework for mitigating inference on VLLMs.
2. The paper is well-written and easy to follow.
3. The experimental setup is sound with a comprehensive range of tasks and baselines.

Weaknesses:

1. The methodological contribution is limited. The contribution is a prompting framework for setting up a verification loop for visual understanding tasks. Though the methodological contribution is limited, the results are compelling.
2. Time complexity: The approach adds a large time and cost overhead for every VLLM inference as running the classification and the verification loop requires multiple LLM calls to an LLM.

---

> ### Author Response · Authors · 2024-06-07
> **Response to Reviewer 3BEp**
>
> Thank you Reviewer 3BEp for valuable feedback.
> > Weakness.  The methodological contribution is limited. The contribution is a prompting framework for setting up a verification loop for visual understanding tasks. Though the methodological contribution is limited, the results are compelling.
>
> Thanks for your insightful comment. In addition to the verification loop, We are the first to distinguish treatment based on the classification of hallucinations and propose a unified framework. We believe applying divide-and-conquer to hallucination mitigation is crucial because it is difficult to solve various hallucinations with a fixed method.
>
> > Weakness.  Time complexity: The approach adds a large time and cost overhead for every VLLM inference as running the classification and the verification loop requires multiple LLM calls to an LLM.
>
> Thanks for your comment. In order to check the time cost of our framework in the experiment, we also recorded the time spent by Dentist and Woodpecker [1] on POPE. The results are as follows: For InstructBLIP, Dentist complete this task with an average of 15.6 seconds per query. Woodpecker complete the task with an average of 16.5 seconds per query. Our method can achieve better performance and less time-consuming compared with the existing the state-of-the-art model.
>
> [1] Woodpecker: Hallucination Correction for Multimodal Large Language Models. CoRR,
>  abs/2310.16045 (2023)
>
> > RC. Could you present an ensemble baseline where the VLLM is called multiple times, and for each generation, we leverage your classification to detect if there are any errors? For evaluation, if all the generations have errors, deem the sample incorrect, else randomly sample one of the generations where neither a perception nor a reasoning error is detected.
>
> Thanks for your suggestion. Following your comment, we have added a baseline called Direct Rejection Baseline (DR Baseline) which means if all the generations of one sample have errors, deem the sample incorrect, else we randomly choose one of the generations which have no errors to be the answer. In addition, we have also provided a new baseline: Repeated Correction Baseline (RC Baseline), which means for each example, when our framework detects hallucinations in all ten responses of the model, the hallucinations are corrected normally, for a more comprehensive analysis.
>
> For evaluation, we take your suggestion and evaluate the baseline and our framework Dentist on POPE. We speculate that the performance of DR Baseline will significantly decrease because only the classification function of our framework are enabled to detect errors, while the answer correction function of our framework is disabled, and the performance of RC Baseline should not show much decline.
>
> Table.1. Results on POPE under the random sampling setting. The experiment setting is the same as the Section 4.1 in our paper.
>
> | LVLM         | Method      | Accuracy | Precision | Recall | F1 Score | Yes Rate |
> | ------------ | ----------- | -------- | --------- | ------ | -------- | -------- |
> | InstrcutBLIP | DR Baseline | 79.17    | 81.81     | 75.00  | 78.26    | 38.33    |
> | InstrcutBLIP | RC Baseline | 85.56    | 92.48     | 82.12  | 81.37    | 40.22    |
> |              |             |          |           |        |          |          |
> | LLaVA        | DR Baseline | 71.70    | 71.93     | 74.55  | 73.21    | 44.34    |
> | LLaVA        | RC Baseline | 87.33    | 86.97     | 91.38  | 82.79    | 46.41    |
> |              |             |          |           |        |          |          |
> | VisualGLM    | DR Baseline | 50.83    | 51.79     | 92.06  | 66.29    | 89.16    |
> | VisualGLM    | RC Baseline | 76.88    | 74.35     | 86.54  | 78.18    | 63.25    |
>
> From the table, we can draw the following conclusions: (1) The results are consistent with our predictions. Using our framework to classify and detect errors without correcting the answers leads to a significant drop in the performance of DR Baseline, while the performance of RC Baseline did not show much decline. DR Baseline show a drop of more than 10% in all metrics, especially the Accuracy of VisualGLM, which drops by 25.05%. (2) This shows that our framework is very effective in detecting  hallucinations generated by models and correcting the model’s answers.
>
> **We add more detailed explanations in Appendix A.1 in the revision.**

---

### Review · Reviewer_H8tC · 2024-06-14

**Summary Of Contributions:**

Paper summary
- The paper proposes Dentist, a framework for mitigating hallucinations in large vision-language models (LVLMs) by iteratively using a LLM (along with the initial LVLM) to refine the initial LVLM response for VQA or image captioning.  Starting from an initial response (answer) from an LVLM for given query and image, The Dentist framework first takes the query and initial response and classifies whether it it likely to generate perception or reasoning hallucinations.  Based on the likely hallucination type, the framework then selects an appropriate strategy (either a perception-oriented strategy or a chain-of-thought prompting for reasoning) to revise the answer to mitigate the hallucination.  The framework iteratively revise the answer until the answer no longer changes significantly (or a maximum number of iterations is reached).  The components of the framework are all built by prompting ChatGPT.  Experiments on MMBench, LLaVA-QA90, CHAIR, POPE using several LVLMs (InstructBLIP, LLaVA, VisualGLM) show that the proposed framework can improve performance over just using the initial response from the LVLMs.  An ablation study was also conducted on MMBench with InstructBLIP.

Main contributions are:
- Dentist, a proposed frame for hallucination mitigation in LVLMS by using LLM to help refine the initial LVLM response
- Set of experiments comparing the proposed framework against the baseline of direct querying of the LVLM, and Woodpecker (prior work that also used LLMs to correct the response).

**Audience:**

Yes

**Broader Impact Concerns:**

No Broader Impact Statement is included.  A Broader Impact Statement is recommended to indicate that mitigating hallucination can help the trustworthiness of the LVLM, but hallucinations can still occur.

**Claims And Evidence:**

No

**Requested Changes:**

- Include missing details, discussion of similarity and differences to Woodpecker, discussion of reproducibility of results, and analysis of what hallucinations are reduced by the proposed framework (some more qualitative examples should be provided at least).
- Writing and presentation improvements.  Below are some writing and presentation issues that have been noted.  These are not comprehensive and meant to just be some examples.
  1. The introduction should be improved to clearly state the tasks being addressed (e.g. VQA and image captioning), and the expected inputs.  It is not clear until section 3 and Figures 2, what the problem setup is is, what are components vs general descriptions, and the inputs and outputs to the framework and its components.

  2. Some sentences are overly long, confusing, or difficult to parse.  A trimming and re-wording pass is recommended to make the writing more straightforward, clear and easier to understand.  Some examples are given here:
     a. "the effect of a method that uses object detection on pictures to verify whether the object in the answer exists will be reduced when the query is the reasoning type" => the clauses in this sentence should be restructured so it is easier to follow.  it's unclear whether this sentence is trying to say "for reasoning queries, it is not effective to use object detection on pictures to verify whether the object in the answer exists" or something else.

     b. "Potential hallucination classification divides the query into two categories: perception and reasoning, which also classifies potential hallucinations caused by these queries"
        - At this point, it is not clear what is meant by a "query" and why that is the appropriate thing to classify (vs using the generated response from the LVLM).  Later in section 3, it becomes more clear what a query is, and from 3.2 (equation 3), it becomes clear that the query and response are used together to determine classification type.
        - What does "which also classifies potential hallucinations caused by these queries" mean?
        - It is also good to either introduce the components before hand ("Potential hallucination classification") or use typesetting (e.g. italics) to indicate that "Potential halluciation classification" is a component in the system (vs just a general concept)

     c. "with a detailed superiority analysis" => superiority analysis does not seem appropriate, please drop this phrase
     d. It is confusing to have statements that refer to experiments on "CHAIR" or "POPE" as CHAIR is a metric and POPE is a evaluation strategy

  3. Provide more concrete examples of prompt and response in the main paper

  4. Tone down inaccurate claims
     - "we propose a unified framework for all kinds of hallucination mitigation" => only two types of hallucination is identified and there are just two types of mitigation strategies
     - "We conduct massive experiments" => As noted, the scale of the evaluation datasets are very small.  The claim of "massive experiments" are justified.

  5. Other writing issues and clarity improvements:
     a. Figure 2 improvements
      - Clearly indicate that the "potential hallucination classification" module takes the answer and query as input (currently it is unclear).
      - The caption should explain what part uses the LVLM being studied and which part uses ChatGPT.
      - The figure can either be made more compact or provide example of actual input / output to different components
     b. Section 3.1: "Noting i" => "Note i" (i should be typeset in math mode), "... and we omit" => it is better to separate into to sentences ("We omit...")
     c. Section 4.1: Use of the word "dimension" for the different levels of MMBench.  The MMBench levels are hierarchical.  The word "dimension" implies separate axis/dimensions.  That is not the case.  It is also confusing to say that "we conduct experiments under the setting of the L-3 level abilities", and for clear to indicate that you "report evaluation for the most fine-grained L-3 level".  As the levels are hierarchical, the same answers that are used to obtain the L-3 evaluation results can also be used to obtain the L-2 and L-1 level results.
     d. Table 1 improvements
       -The highest model should be consistently bolded, not just when it is Dentist.  Stylistically, I would recommend removing the % and reducing the number of horizontal lines (only the horizontal line before Overall is needed).
       - Should include Woodpecker for comparison
       - It would be better if the L-3 abiltiies are grouped by the hierarchy (as originally intended by MMBench) vs alphabetically.  It's also unclear if it is necessarily to have the L-3 granularity for the results.  L-2 granularity would be sufficient

     e. Section 4.1: "we also introduce the baseline Woodpecker" => "we compare against the baseline Woodpecker" (Woodpecker is prior work from Yin et al. 2023, and not introduced by this work)
     Citation for GPT-3.5-turbo should not be Brown et al. 2020, as it is built with additional work that follows Brown et al. 2020. and the specific version used should be specified.

     d. Table 4. Please add horizontal lines grouping the LVLMs.  The bolding should be per LVLM.
     e. Figure 4. Improve caption to provide more details about the two examples.
     f. Minor: Section 2.2 "hires an LLM" => "uses an LLM"
- Include Discussion of recent relevant work. Some examples (not comprehensive) are given
  1. Hallucination mitigation for LVLMs.
  - Mitigating Object Hallucinations in Large Vision-Language Models through Visual Contrastive Decoding [Leng et al. CVPR 2024]
  - Opera: Alleviating hallucination in multi-modal large language models via over-trust penalty and retrospection-allocation [Huang et al. CVPR 2024]
  - RLHF-V: Towards trustworthy MLLMs via behavior alignment from fine-grained correctional human feedback [Yu et al. CVPR 2024]
  - Hallucination of Multimodal Large Language Models: A Survey [Bai et al. 2024] can potentially be good reference for relevant work
  2. Using LLMs to improve task performance of VLMs
  - Filling the Image Information Gap for VQA: Prompting Large Language Models to Proactively Ask Questions [Wang et al. 2023]
  - Investigating Prompting Techniques for Zero- and Few-Shot Visual Question Answering [Awal et al. 2024]

**Strengths And Weaknesses:**

Strengths
- Hallucination is an important problem to address
- The proposed solution is straightforward and does not require additional training or fine-tuning
- Results show that the proposed method can improve performance over the baseline of querying the LVLM directly

Weaknesses
- Writing and presentation is poor and is inaccurate and confusing at places (see requested changes for details)
- Not enough discussion of how Woodpecker is similar to and differs from the proposed framework
- Some details of the method is missing and evaluation setup are missing
  - Specifics of the GPT-3.5-turbo model used and cost is not specified.
  - How is the Similar function in Algorithm 1 implemented?
  - For MMBench, was the evaluation done with VanillaEval or CircularEval, on MMBench-dev or MMBench-test?
  - Limited statistics are provided for each dataset that are used for evaluation and scattered and difficult to find.  A table will help. How large was MMBench?  It should be make clear that LLaVA-QA90 has "90 questions" not "90 VQA tasks".
- Lack of analysis on what specific hallucinations are actually reduced by the proposed method.
- No discussion of how reproducible the results are
  - Will the LLM / LVLM give different responses when prompted again?
  - The datasets used for evaluation are very small and it is unclear whether the experimental results are robust.  For instance, LLaVA-QA90 only has 90 questions over 30 images, and the CHAIR and POPE setting are also very small (50 images for CHAIR, 100 images with 6 questions for each sampling setting).  Can we expect results on such small datasets to be statistically meaningful and robust across runs?
- Missing discussion of recent work on halluciation mitigtation in LVLMs (see requested changes)

---

> ### Author Response · Authors · 2024-07-14
> **Response to Reviewer H8tC**
>
> Thank you for valuable feedback. We addressed your concerns as follows.
>
> > Weakness.
> >
> > - Some details of the method is missing and evaluation setup are missing
> >     - Specifics of the GPT-3.5-turbo model used and cost is not specified.
> >     - How is the Similar function in Algorithm 1 implemented?
> >     - For MMBench, was the evaluation done with VanillaEval or CircularEval, on MMBench-dev or MMBench-test?
> >     - Limited statistics are provided for each dataset that are used for evaluation and scattered and difficult to find. A table will help. How large was MMBench? It should be make clear that LLaVA-QA90 has "90 questions" not "90 VQA tasks".
> 1. We utilize gpt-3.5-turbo-0613. We add the detail of GPT in Section 4.1. The cost for calling its API is 1.5 USD / 1M tokens for input and 2.0 USD / 1M tokens for output. We calculated the cost of calling the gpt-3.5-turbo-0613 API when evaluating on MMBench. In one round of evaluation, the total cost was about 2.75 USD, with an average cost of 0.0004 USD per question. We believe this cost is acceptable. We add these content in Appendix A.13.
> 2. The Similar function in Algorithm 1 is implemented through the prompt template in Appendix A.7. We ask GPT-3.5 to identify whether there is a conflict between answer from the previous round and answer from this round. We clarify it in Section 3.4.
> 3. MMBench evaluation was done with CircularEval on MMBench-Test (EN). We supplemented this in Section 4.1.
> 4. The MMBench dataset consists of 6,666 samples, each of which contains a question, an image and several optional answers.
>    We add this table in Appendix A.11.
>
>
>     | Benchmarks | Questions | Tasks |
>     | --- | --- | --- |
>     | LlaVA-QA90 | 90 | generative |
>     | POPE | 1,800 | classification |
>     | MMBench | 6,666  | classification |
>     | CHAIR | 2,000 (extended in the revision) | generative |
>
> > Weakness.
> >
> > - No discussion of how reproducible the results are
> >     - Will the LLM / LVLM give different responses when prompted again?
> >     - The datasets used for evaluation are very small and it is unclear whether the experimental results are robust. For instance, LLaVA-QA90 only has 90 questions over 30 images, and the CHAIR and POPE setting are also very small (50 images for CHAIR, 100 images with 6 questions for each sampling setting). Can we expect results on such small datasets to be statistically meaningful and robust across runs?
> 1. When the model parameters are fixed, the answer is the same. The parameters can refer to Appendix A.14. And we add the discussion on reproducibility in Appendix A.2.
> 2. Regarding the dataset setting:
>    (1) In the LlaVA-QA90 evaluation, we followed the settings of previous work (such as Visual Instruction Tuning [Liu, Haotian et al. 2023] ), so we believe that this setting is reasonable.
>    (2) In the POPE evaluation, we sampled 100 pictures. Since each picture corresponds to three sampling settings, each sampling setting contains 6 questions, so a total of 1,800 questions are included. Compared with previous work (such as Woodpecker sampled 50 pictures), this dataset setting is large enough.
>    (3) As we mentioned above, MMBench contains a total of 6,666 questions.
>    (4) Regarding CHAIR, we previously sampled only 50 images, which is indeed insufficient. In order to further demonstrate the robustness of the framework and the results are statistically meaningful, we expanded the dataset to 2,000  examples. The supplementary experimental results are as follows:
>
>
>     |  | InstructBLIP($C_s$) | InstructBLIP($C_i$) | LLaVA($C_s$) | LLaVA($C_i$) | VisualGLM($C_s$) | VisualGLM($C_i$) |
>     | --- | --- | --- | --- | --- | --- | --- |
>     | Baseline | 59.2 | 22.9 | 84.0 | 23.4 | 44.0 | 17.8 |
>     | Dentist | 52.1 | 16.7 | 75.2 | 17.1 | 36.0 | 11.3 |
>     | Woodpecker | 56.9 | 18.5 | 71.9 | 19.8 | 37.0 | 11.9 |
>
>     It can be seen that after expanding the CHAIR dataset, the test results are slightly different, but the results are consistent. Therefore, we believe that our experimental results are statistically meaningful.
>
>     We modify the results about CHAIR in Table 3 in the revision.
>
>     [1]Haotian Liu, Chunyuan Li, Qingyang Wu, and Yong Jae Lee. Visual instruction tuning.

---

> ### Author Response · Authors · 2024-07-14
> **Response to Reviewer H8tC**
>
> > **Requested Changes:**
> >
> > - **Include missing details, discussion of similarity and differences to Woodpecker, discussion of reproducibility of results, and analysis of what hallucinations are reduced by the proposed framework (some more qualitative examples should be provided at least).**
> 1. **Missing details:**
>
>     As mentioned in the reply about weakness above, we add the missing details in the corresponding sections in the revision.
>
> 2. **Discussion of similarity and differences to Woodpecker:**
>
>     The similarity between our framework Dentist and Woodpecker is that we both use an LLM to revise the hallucinated response generated by LVLMs.
>
>     Woodpecker extracts main objects from responses and verifies these objects with object segmentation tool and VQA models, for all hallucinated responses. However, Dentist has a divide-and-conquer treatment. When Dentist is faced with perception query, it tries to verify the main objects in the model response. When faced with reasoning query, Dentist uses CoT to deal with it. This divide-and-conquer treatment accurately mitigates hallucinations and achieves better results.
>
>     We added the above comparative analysis to the Section 4.2 in the revision.
>
> 3. **Discussion on reproducibility:**
>
>     We supplemented the relevant experiments and discussed the reproducibility in Appendix A.2 in the revision.
>
>     We focus on the following questions: (1) When LVLM repeatedly generates captions for the same image, will it produce the same hallucinations? (2) When using our framework to process a series of model responses in (1), can we obtain consistent results? Can we guarantee that the hallucinations can be corrected every time? It should be emphasized that since our method is training-free, all parameters of the model are fixed. In this case, LVLM usually produces the same response to the same question, so hallucinations are repeated. Correspondingly, Dentist can also effectively eliminate these repeated hallucinations.
>
>     Please refer to the revised Appendix A.2 for more details.
>
> 4. **Analysis of what hallucinations are reduced:**
>
>     We provided examples of hallucination correction for perception and reasoning in Section 4.4 called Case Study, along with detailed analysis.
>
>     We added cases of mitigating hallucinations in Appendix A.12.
>     ****
>
>
> > **Requested Changes:**
> >
> > - **Writing and presentation improvements.**
>
> Thank you for your valuable comments on the paper writing. We have reviewed and revised the content of our paper based on your comments. We submit the changes in the revision.
>
> > **Requested Changes:**
> >
> > - **Include Discussion of recent relevant work. Some examples (not comprehensive) are given**
>
> Thank you for these examples of recent relevant work.  We read these papers and add the following papers to the related work:
>
> (1) RLHF-V: Towards trustworthy MLLMs via behavior alignment from fine-grained correctional human feedback [Yu et al. CVPR 2024]. We add it to the first category in Section 2.2.
>
> (2) Mitigating Object Hallucinations in Large Vision-Language Models through Visual Contrastive Decoding [Leng et al. CVPR 2024]. We add it to the second category in Section 2.2.
>
> (3) Opera: Alleviating hallucination in multi-modal large language models via over-trust penalty and retrospection-allocation [Huang et al. CVPR 2024]. We add it in Section 2.2.
>
> (4) Hallucination of Multimodal Large Language Models: A Survey [Bai et al. 2024]. We add it in Section 2.2.
>
> (5) Filling the Image Information Gap for VQA: Prompting Large Language Models to Proactively Ask Questions [Wang et al. 2023]. We add it in Section 2.1.
>
> (6) Investigating Prompting Techniques for Zero- and Few-Shot Visual Question Answering [Awal et al. 2024]. We add it in Section 2.1.

---

### Review · Reviewer_e5uA · 2024-08-10

**Summary Of Contributions:**

The paper addresses mitigating hallucinations in Large Visoin-Language Models (LVLMs), a critical issue hampering the performance and reliability of LVLMs. The authors observe that there have been no approaches using different mitigation methods for different types of hallucinations and using a iterative method to fully eliminate the hallucinations in the model's output. Based on this, the paper proposes a new unified framework, Dentist, which first classifies the input query into "reasoning" and "perception", and mitigates hallucinations using different methods for different types of queries in an iterative manner. The experimental results show the superiority of Dentist on several benchmarks, and the ablation study proves the validity of design choices.

**Audience:**

Yes

**Broader Impact Concerns:**

Please add the Broader Impact Statement in the final draft.

**Claims And Evidence:**

Yes

**Requested Changes:**

Please answer the above weaknesses. Also I have minor questions/recommendations:
- The prompt template in Figure 5 does not use model responses, Y hat, while Equation 3 uses them. Is it intended?
- It seems that the precision boost in LLaVA by Dentist on POPE, random setting is 16.92% points, not 14.83% (Section 4.2)
- In Section 4.3, it would be great to provide the exact numbers of misclassification by GPT-3.5-turbo, not just mentioning "We speculate that this is due to the misjudgment by GPT-3.5-turbo ...".
- How many iterations did the authors use for the experiments reported in Section 4.1 (T in Algorithm 1).
- I don't understand the Direct Rejection Baseline (DR Baseline): what do the authors mean by “the sample is directly judged as an error.”? How are they treated when computing precision and recall?

And please add a new section addressing the limitations of the proposed method and future work.

**Strengths And Weaknesses:**

Strengths
- I like the idea of applying divide-and-conquer to hallucination mitigation. As the authors mentioned in the paper, there are various types of hallucinations and we cannot solve them with a single fixed method unless the framework internally processes hallucinations differently.
- The authors provide all the details of the proposed framework, including the prompt templates used for the proposed framework, Dentist.
- Overall, the paper is well-written and easy-to-read.
- The superior experimental results show the validity of Dentist.

Weaknesses: The quality of this work is overall good, but I have some concerns about the proposed framework and experiment designs:
- In Section 3.3, the sub-questions are generated based on the model’s outputs. Thus, if the model does not describe salient features in the image at first, the proposed method cannot guide the model to capture them until the end.
- The prompt template in Figure 10 for evaluating on LLaVA-QA90 needs two passages as input. How did the authors pair among the baseline LVLMs, Dentist and Woodpecker for getting the scores in Table 2? And why did the authors prompt ChatGPT to compare two passages and score them at once? Have the authors prompt ChatGPT to score a single passage from each of the models?
- (Figure 10 and Table 2) For evaluating the rationality of inference content, the authors should also have checked the precision, that is the proportion of logical reasoning sentences included in the passage without hallucination, not just measuring the number of logical reasoning s without hallucinations (recall). Also, I think giving higher scores to the passages with more logical reasoning is quite advantageous to the methods that generate the whole reasoning process like CoT. Thus, I am not sure whether comparison with WoodPecker is fair using this metric because it does not use CoT. Lastly, regardless of whether generating reasoning process or not, I believe that the result of reasoning should be evaluated.
- Please report the performance of WoodPecker on MMbench if possible. It would also be great to report the performance of other previous approaches on the benchmarks.

---

> ### Author Response · Authors · 2024-08-18
> **Response to Reviewer e5uA**
>
> Thank you for valuable feedback. We addressed your concerns as follows.
>
> > Weakness.
> >
> >
> > • In Section 3.3, the sub-questions are generated based on the model’s outputs. Thus, if the model does not describe salient features in the image at first, the proposed method cannot guide the model to capture them until the end.
>
> Thanks for pointing this out. This does happen, but we mainly focus on hallucinations (conflicts between model‘s answers and image content) rather than sailent features of the image. Therefore, our original intention of generating sub-questions based on model’s outputs is to verify possible errors rather than omissions of the original image in model’s answers.
>
>
>
> > Weakness.
> > • The prompt template in Figure 10 for evaluating on LLaVA-QA90 needs two passages as input. How did the authors pair among the baseline LVLMs, Dentist and Woodpecker for getting the scores in Table 2? And why did the authors prompt ChatGPT to compare two passages and score them at once? Have the authors prompt ChatGPT to score a single passage from each of the models?
>
> Thanks for your comments. We answer your comments as follows,
>
> 1. We respectively paired baseline LVLMs with Dentist, baseline LVLMs with Woodpecker [1], and provided their responses to ChatGPT for scoring.  We add this explanation in Section 4.1.
> 2. The reason why ChatGPT is prompted to compare two paragraphs and score them at once is  to ensure that the scores are mutually referenced, thereby making them more reliable. We designed this experiment and prompt following Woodpecker. We add relevant explanation in Section 4.1.
> 3. We didn’t prompt ChatGPT to score a single passage. This is because scoring a single passage may lead to significant differences in repeated experiments.
>
> [1] Woodpecker: Hallucination Correction for Multimodal Large Language Models. CoRR, abs/2310.16045 (2023)
>
> > Weakness.
> > • (Figure 10 and Table 2) For evaluating the rationality of inference content, the authors should also have checked the precision, that is the proportion of logical reasoning sentences included in the passage without hallucination, not just measuring the number of logical reasoning s without hallucinations (recall). Also, I think giving higher scores to the passages with more logical reasoning is quite advantageous to the methods that generate the whole reasoning process like CoT. Thus, I am not sure whether comparison with WoodPecker is fair using this metric because it does not use CoT. Lastly, regardless of whether generating reasoning process or not, I believe that the result of reasoning should be evaluated.
>
> Thanks for your insightful suggestions and we response as follows,
>
> 1. Following for advice, we compute the precision, that is the proportion of logical reasoning sentences included in the passage without hallucination. The results are shown as follows.
>
>
>    |            | InstructBLIP | LLaVA  | VisualGLM |
>    | ---------- | ------------ | ------ | --------- |
>    | Baseline   | 0.2103       | 0.2727 | 0.3647    |
>    | Woodpecker | 0.2096       | 0.2656 | 0.3827    |
>    | Dentist    | 0.2281       | 0.2795 | 0.3976    |
>
>
> 2. The logicality metric may favor methods that use CoT. But this metric is just one facet of our evaluation. We also perform well in Accuracy and Detailedness, which indicates that Dentist is not only superior in handling reasoning queries compared to previous methods, but also excels in the accuracy and meticulousness of describing the content in images.
> 3. Indeed, the results of reasoning should be evaluated. We have prompted ChatGPT to score reasonable reasoning processes and results which do not contains hallucinations higher on Logicality, so both the reasoning process and results are under our consideration.

---

> ### Author Response · Authors · 2024-08-18
> **Response to Reviewer e5uA**
>
> > Weakness.
> >
> >
> > • Please report the performance of WoodPecker on MMbench if possible. It would also be great to report the performance of other previous approaches on the benchmarks.
>
> Thanks for your comment. The following are the results of Woodpecker on MMBench. We supplemented the performance of Woodpecker on MMBench in Table 1, Section 4.2 in the revision. From the results, it can be seen that Dentist performs better than Woodpecker in the vast majority of abilities.
>
> |                       | InstructBLIP-7B (Dentist) | InstructBLIP-7B (Woodpecker) | LLaVA-V1.5-7B (Dentist) | LLaVA-V1.5-7B (Woodpecker) | VisualGLM-6B (Dentist) | VisualGLM-6B (Woodpecker) |
> | --------------------- | ------------------------- | ---------------------------- | ----------------------- | -------------------------- | ---------------------- | ------------------------- |
> | Image Topic           | 60.0                      | 43.5                         | 96.4                    | 87.1                       | 50.2                   | 64.7                      |
> | Image Quality         | 10.6                      | 0.0                          | 10.2                    | 0.0                        | 15.6                   | 3.5                       |
> | Image Emotion         | 41.3                      | 12.0                         | 76.4                    | 67.5                       | 50.6                   | 33.7                      |
> | Image Scene           | 60.3                      | 57.4                         | 88.3                    | 79.8                       | 70.3                   | 59.7                      |
> | Image Style           | 37.1                      | 38.8                         | 55.3                    | 58.8                       | 35.8                   | 24.7                      |
> | OCR                   | 58.6                      | 36.4                         | 78.3                    | 67.5                       | 43.6                   | 53.2                      |
> | Celebrity Recognition | 49.2                      | 68.6                         | 68.6                    | 60.2                       | 55.2                   | 46.6                      |
> | Object Localization   | 14.4                      | 8.6                          | 11.5                    | 14.4                       | 10.9                   | 8.6                       |
> | Attribute Recognition | 51.4                      | 52.5                         | 70.6                    | 66.7                       | 43.7                   | 33.3                      |
> | Action Recognition    | 57.5                      | 20.4                         | 85.2                    | 80.7                       | 38.6                   | 28.4                      |
> | Attribute Comparison  | 2.6                       | 0.0                          | 25.8                    | 6.4                        | 10.8                   | 6.4                       |
> | Spatial Relationship  | 8.6                       | 11.1                         | 15.3                    | 11.1                       | 10.9                   | 8.6                       |
> | Identity Reasoning    | 68.3                      | 70.7                         | 86.6                    | 86.6                       | 88.4                   | 71.2                      |
> | Function Reasoning    | 49.6                      | 50.9                         | 77.8                    | 73.6                       | 50.6                   | 37.7                      |
> | Physical Property     | 21.9                      | 26.0                         | 50.0                    | 53.0                       | 30.3                   | 17.0                      |
> | Nature Relation       | 27.3                      | 24.7                         | 38.3                    | 38.3                       | 30.6                   | 8.6                       |
> | Physical Relation     | 17.3                      | 19.2                         | 28.9                    | 26.9                       | 9.6                    | 3.8                       |
> | Social  Relation      | 41.0                      | 38.4                         | 69.6                    | 62.8                       | 45.3                   | 17.9                      |
> | Image-Text            | 7.0                       | 15.8                         | 10.9                    | 12.9                       | 5.0                    | 8.9                       |
> | Future Prediction     | 55.0                      | 25.0                         | 52.1                    | 44.4                       | 31.0                   | 6.9                       |

---

> ### Author Response · Authors · 2024-08-18
> **Response to Reviewer e5uA**
>
> > Requested Changes.
> >
> >
> > • The prompt template in Figure 5 does not use model responses, Y hat, while Equation 3 uses them. Is it intended?
>
> Thank you for pointing out our errata. We modify the Equation 3 in Section 3.2.
>
> > Requested Changes.
> >
> >
> > • It seems that the precision boost in LLaVA by Dentist on POPE, random setting is 16.92% points, not 14.83% (Section 4.2)
>
> Thank you for pointing out this issue. The correct description is: Dentist boosts the precision of LLaVA by 16.92% in the popular setting. We have reviewed and corrected it in Section 4.2.
>
> > Requested Changes.
> >
> >
> > •  In Section 4.3, it would be great to provide the exact numbers of misclassification by GPT-3.5-turbo, not just mentioning "We speculate that this is due to the misjudgment by GPT-3.5-turbo ...".
>
> Thank you for your suggestion. To illustrate this issue, we extract 1000 samples from the test results of MMBench and manually count the number of classification errors of Dentist. Among these 1,000 samples, 33 queries are misclassified, with an error rate of approximately 3.3%. Compared to its improvement in LVLMs performance, we believe this quantity is acceptable since our method has achieved state-of-the-art performance on several benchmarks. We added this result in Section 4.3 in the revision.
>
> > Requested Changes.
> >
> >
> > • How many iterations did the authors use for the experiments reported in Section 4.1 (T in Algorithm 1).
>
> In the experiment mentioned in Section 4.1, Dentist iterates at most three times in the verification cycle to ensure the effectiveness of the verification and avoid excessive time costs. We added this explanation in Section 4.1.
>
> > Requested Changes.
> >
> >
> > • I don't understand the Direct Rejection Baseline (DR Baseline): what do the authors mean by “the sample is directly judged as an error.”? How are they treated when computing precision and recall?
>
> Thanks for your suggestions. This baseline is based on the comments of Reviewer 3BEp. Specifically, for each sample we call LVLM ten times  to generate multiple responses. "the sample is directly judged as an error." means  if all the responses have errors, deem the sample incorrect. For a incorrect sample,  when calculating precision and recall,  if it is initially considered positive sample, we consider it as a false positive sample; On the contrary, consider it as a false negative sample.
>
> > Requested Changes.
> >
> >
> > • And please add a new section addressing the limitations of the proposed method and future work.
>
> Thank you for your suggestion. We add Section 6 to discuss the limitations and possible future work in the revision.

---

> > ### Comment · Reviewer_e5uA · 2024-08-22
> >
> > Thank you for your detailed responses. Most of my concerns are addressed, but I have a few comments.
> >
> > - It seems that baseline LVLMs are scored two times: with Dentist and with Woodpecker, but only a single score is reported in Table 2. Did you average the scores?
> > - Accuracy and Detailedness seem to be more related to perception, and that's why I requested the precision score. Please add the precision score in the final draft.

---

> > > ### Author Response · Authors · 2024-08-23
> > > **Response to Reviewer e5uA**
> > >
> > > Thank you for your valuable comments.
> > >
> > > > • It seems that baseline LVLMs are scored two times: with Dentist and with Woodpecker, but only a single score is reported in Table 2. Did you average the scores?
> > >
> > > Actually we rely on these two scores of the baseline to align the scores of Dentist and Woodpecker. Specifically, if the two sets of scores are as follows :
> > > |         | Score            |
> > > | ------- | ---------------- |
> > > | LVLMs   | $S_{Baseline-1}$ |
> > > | Dentist | $S_{Dentist}$    |
> > >
> > > |            | Score            |
> > > | ---------- | ---------------- |
> > > | LVLMs      | $S_{Baseline-2}$ |
> > > | Woodpecker | $S_{Woodpecker}$ |
> > >
> > > Then the last aligned scores are as follows :
> > >
> > > |            | Score                                                      |
> > > | ---------- | ---------------------------------------------------------- |
> > > | LVLMs      | $S_{Baseline-1}$                                           |
> > > | Dentist    | $S_{Dentist}$                                              |
> > > | Woodpecker | $S_{Woodpecker} \times S_{Baseline-1} \div S_{Baseline-2}$ |
> > >
> > > The purpose of the above calculation is to normalize the scores of Dentist and Woodpecker to the same scale.
> > >
> > > > • Accuracy and Detailedness seem to be more related to perception, and that's why I requested the precision score. Please add the precision score in the final draft.
> > >
> > > Thank you for your advice. We add the Precision in Table 2 in Section 4.2.

---

### Comment · Action_Editor_Znak · 2024-07-14
**Apologies for the delay**

Dear all,

Apologies for the delay in releasing the reviews to all reviewers.

We are still waiting for one reviewer to submit their review (and I have assigned additional reviewers, given that this reviewer has not responded to reminders or emails either). This is unfortunate and is causing quite a delay: even when a new reviewer acknowledges the review assignment, it will take a bit given that this is a long paper.

Best wishes,\
AE

---

### Decision · Action_Editor_Znak · 2024-09-18

**Recommendation:** Accept with minor revision

**Comment:**

Thank you to the reviewers for their reviews and improvement suggestions, and thank you to the authors for their patience with the delays during the summer break.

Overall, the decision is to accept the paper with minor revisions.

This paper presents a framework for mitigating hallucinations in large vision-language models (LVLMs) through an iterative prompting approach. The key idea is to classify queries as either perception or reasoning related and then apply targeted iterated strategies to refine the model's output. The authors evaluate their approach on several benchmarks and demonstrate improvements over baseline LVLMs and previous hallucination correction methods. The authors have addressed concerns raised by the reviewers in their revision, including clarifying missing details on their experimental setup and expanding their empirical evaluation and literature review, as well as fixing writing issues and improving the presentation.

The following minor revisions are recommended/needed based on comments from the reviewer H8tC's recommendations (which are not visible to the authors):

1. I'm not sure whether the final "Response to Reviewer e5uA" clarifying how the scoring of the baseline LVLM for two methods (Dentist and Woodpecker) are aligned is incorporated into the manuscript. It would be good if that was included.

2. In addition, some parts of the manuscript is unclear (due to wording). There should be a pass over the text to improve the wording. Some suggestions are given below. A more complete proofreading pass to improve wording is recommended.
Writing improvements

3. Please reword to be more accurate:
  - Page 13: "To evaluate the effectiveness of our framework, We conduct massive experiments" => "To evaluate the effectiveness of our framework, we conduct experiments"
4. Please reword to be clearer:
  - Page 6: "Regarding automatically determining whether the answer is no longer changing semantically, we use a prompt to make ChatGPT automatically complete it instead of manual work. In Algorithm 1, the Similar function represents this prompt. Refer to Appendix A.7 for the prompt template." => Please reword to be clearer. For example: "We use ChatGPT to determine whether the answer has converged and is no longer changing semantically. This corresponds to the Similar function in Algorithm 1 and the prompt template is given in Appendix A.7".
  - Page 8: "In Section 3.4, we mention that Dentist adopt validation loops." => cut or reword to be clearer
  - Page 9: "promotes the detail of image description and the rationality of inference content" => please reword
  - Page 9: "can increase effective reasoning content" => please reword
5. Logically, it would be better if "Limitations And Future Work" (Section 6) comes before "Conclusion" (Section 5)
6. Other recommended text rewordings:
  - Page 4: Figure 2 caption: "The orange square components use gpt." => "The components using GPT are indicated in orange."
  - Page 7: "CHAIRi" => "CHAIR_i"
  - Page 8: "Dentristry" => "Dentist" \
    "Limitations And Future Work" => "Limitations and Future Work" \
    "Discussion On Reproductivity" => "Discussion on Reproducibility"

Additionally, the following comments from my own reading of the current revision:
1. " any part of the answer that does not match the content of the picture will be corrected by our framework Dentist." => This is too strong a claim, please tone down.
2. "widely used method dedicated to evaluating" => "evaluation method"
3. "w/o-Per/Rea/Cla" => please use clearer names for the ablations - the table is narrow currently and has sufficient horizontal space to improve readability
4. The authors might want to more explicitly clarify that their verification loop is a fixed point iteration and maybe refer to the theory as future work?
4. Why "dentist"? The paper does not explain the name at all (except for a somewhat strange metaphor). It might be worth choosing a meaningful acronym.
5. In general, edit phrases to be more concise and plain and thus avoid unclear wording. This will significantly help non-native readers.

Congratulations on the acceptance, and I'm looking forward to the camera-ready revision.

Best wishes,\
Andreas

**Audience:**

This paper should interest researchers working on large vision-language models, particularly those focused on improving model reliability and reducing hallucinations. Hallucination mitigation is an essential challenge, and the proposed framework offers a simple but effective approach that could be widely applicable.

**Claims And Evidence:**

The authors have conducted experiments on multiple benchmarks (MMBench, POPE, CHAIR, LLaVA-QA90) to demonstrate the effectiveness of their proposed framework (Dentist) for mitigating hallucinations in large vision-language models. They have also provided ablation studies and comparisons to relevant baselines. The claims made in the submission are generally evidenced. The reviewers acknowledged the improvements the authors made during the reviewing process.

---

> ### Author Response · Authors · 2024-09-24
> **Update Camera Ready Version**
>
> Dear Action Editor Znak,
>
> Thank you very much for appreciating our work. We have revised the paper with the following changes:
>
> - We corrected the existing writting issues based on the suggestions of AE and reviewers.
>
> - We made some additions based on the suggestions. For example, we clarify how the scoring of the baseline LVLM for two methods (Dentist and Woodpecker) are aligned in Appendix A.15.
>
> Again, thank you very much for your attention and support, and wish you a good day.
>
> Best regards,
>
> Paper 2644 authors